# Generalized Boosting

**Arun Sai Suggala, Bingbin Liu, Pradeep Ravikumar**
Carnegie Mellon University
Pittsburgh, PA 15213
{asuggala,bingbinl,pradeepr}@cs.cmu.edu

## Abstract

Boosting is a widely used learning technique in machine learning for solving classification problems. In boosting, one predicts the label of an example using an ensemble of weak classifiers. While boosting has shown tremendous success on many classification problems involving tabular data, it performs poorly on complex classification tasks involving low-level features such as image classification tasks. This drawback stems from the fact that boosting builds an additive model of weak classifiers, each of which has very little predictive power. Often, the resulting additive models are not powerful enough to approximate the complex decision boundaries of real-world classification problems. In this work, we present a general framework for boosting where, similar to traditional boosting, we aim to boost the performance of a weak learner and transform it into a strong learner. However, unlike traditional boosting, our framework allows for more complex forms of aggregation of weak learners. In this work, we specifically focus on one form of aggregation - *function composition*. We show that many popular greedy algorithms for learning deep neural networks (DNNs) can be derived from our framework using function compositions for aggregation. Moreover, we identify the drawbacks of these greedy algorithms and propose new algorithms that fix these issues. Using thorough empirical evaluation, we show that our learning algorithms have superior performance over traditional additive boosting algorithms, as well as existing greedy learning techniques for DNNs. An important feature of our algorithms is that they come with strong theoretical guarantees.

## 1 Introduction

Boosting is a widely used learning technique in machine learning for solving classification problems. Boosting aims to improve the performance of a weak learner by combining multiple weak classifiers to produce a strong classifier with good predictive performance. Since the seminal works of Freund [13], Schapire [32], a number of practical algorithms such as AdaBoost [16], gradient boosting [26], XGBoost [9], have been proposed for boosting. Over the years, boosting based methods such as XGBoost in particular, have shown tremendous success in many real-world classification problems, as well as competitive settings such as Kaggle competitions. However, this success is mostly limited to classification tasks involving structured or tabular data with hand-engineered features. On classification problems involving low-level features and complex decision boundaries, boosting tends to perform poorly [3, 30] (also see Section 5). One example where this is evident is the image classification task, where the decision boundaries are often complex and the features are low-level pixel intensities. This drawback stems from the fact that boosting builds an additive model of weak classifiers, each of which has very little predictive power. Since such additive models with any reasonable number of weak classifiers are usually not powerful enough to approximate complex decision boundaries, the models' output by boosting tend to have poor performance.

In this work, we aim to overcome this drawback of traditional boosting by considering a generalization of boosting which allows for more complex forms of aggregation than linear combinations of weak classifiers. To achieve this goal, we work in the feature representation space and boost the

performance of *weak feature transformers*. Working in the representation space allows for more flexible combinations of weak feature transformers. This is unlike traditional boosting which works in the label space and builds an additive model on the predictions of the weak classifiers. The starting point for our approach is the greedy view of boosting, originally studied by Friedman et al. [18], Mason et al. [26]. Letting $\widehat{R}_S(f)$ be the risk of a classifier $f$ on training samples $S$, boosting techniques aim to approximate the minimizer of $\widehat{R}_S$ in terms of linear combinations of elements from a set of weak classifiers $\mathcal{F}$. Many popular boosting algorithms including AdaBoost, XGBoost, rely on greedy techniques to find such an approximation. In our generalized framework for boosting, we take this greedy view, but differ in how we aggregate the weak learners. We approximate the minimizer of $\widehat{R}_S$ using models of the form $f_T = W\phi_T$, where $\phi_T = \sum_{t=0}^T g_t$, and $\{g_t\}_{t=0}^T$ are feature transformations learned in each iteration of the greedy algorithm, and $W$ is the linear classifier on top of the feature transformation. Unlike additive boosting, where each $g_t$ comes from a fixed weak feature transformer class $\mathcal{G}$, in our framework each $g_t$ comes from a class $\mathcal{G}_t$ which evolves over time $t$ and is allowed to depend on the past iterates $\{\phi_i\}_{i=0}^{t-1}$. Some potential choices for $\mathcal{G}_t$ that could be of interest are $\{g \circ \phi_{t-1} \text{ for } g \in \mathcal{G}\}$, $\{g \circ ([\phi_0, \ldots, \phi_{t-1}]) \text{ for } g \in \mathcal{G}\}$, where $g \circ \phi(\mathbf{x}) = g(\phi(\mathbf{x}))$ denotes function composition of $g$ and $\phi$, and $\mathcal{G}$ is a weak feature transformer class. Note that the former choice of $\mathcal{G}_t$ is connected to layer-by-layer training of models with ResNet architecture [21].

As one particular instantiation of our framework, we consider weak feature transformers that are neural networks and use function compositions to combine them; that is, we use $\mathcal{G}_t$'s constructed using function compositions. We show that for certain choices of $\mathcal{G}_t$, our framework recovers the layer-by-layer training techniques developed in deep learning [6, 22]. Greedy layer-by-layer training techniques have seen a revival in recent years [5, 8, 22, 25, 29]. One reason for this revival is that greedy techniques consume less memory than end-to-end training of deep networks, as they do not perform end-to-end back-propagation. Consequently, they can accommodate much larger models in limited memory. As a primary contribution of the paper, we identify several drawbacks of existing layer-by-layer training techniques, and show that the choice of $\mathcal{G}_t$ used by these algorithms can lead to a drop in performance. We propose alternative choices for $\mathcal{G}_t$ which fix these issues and empirically demonstrate that the resulting algorithms have superior performance over existing layer-by-layer training techniques, and in some cases achieve performance close to that of end-to-end trained DNNs. Moreover, we show that the proposed algorithms perform much better than traditional additive boosting algorithms, on a variety of classification tasks.

As the second contribution of the paper, we provide excess risk bounds for models learned using our generalized boosting framework. Our results depend on a certain weak learning condition on feature transformer classes $\{\mathcal{G}_t\}_{t=1}^T$, which is a natural generalization of the weak learning condition that is typically imposed in traditional boosting. The resulting risk bounds are modular and depend on the generalization bounds of $\{\mathcal{G}_t\}_{t=1}^T$. An advantage of such modular bounds is that one can rely on the best-known generalization bounds for weak transformation classes $\{\mathcal{G}_t\}_{t=1}^T$ and obtain tight risk bounds for boosting. As an immediate consequence of this result, we obtain excess risk bounds for existing greedy layer-by-layer training techniques.

**Related Work.** Several works have proposed generalizations of traditional boosting [10, 11, 20, 22]. Cortes et al. [10] propose a boosting algorithm where the hypothesis set of weak classifiers is chosen adaptively. However, the resulting models are still additive models of weak classifiers and usually perform poorly on hard classification problems. Several recent works have attempted to learn neural networks greedily based on boosting theory. Cortes et al. [11] propose a boosting-style algorithm to learn both the structure and weights of neural networks in an adaptive way. However, the algorithms developed are restricted to feed forward neural networks and are mostly theoretical in nature. The experimental evidence in the paper is a proof-of-concept and only considers small scale binary classification tasks. Huang et al. [22], Nitanda and Suzuki [29] use ideas from classical boosting to learn neural networks in a layer-by-layer fashion. As we show later, these algorithms are specific instances of our generalized framework, and have certain drawbacks arising from the choice of $\mathcal{G}_t$ they use.

## 2 Preliminaries

In this section, we set up the notation and review the necessary background on additive boosting. A consolidated list of notation can be found in Appendix A.

**Notation.** Let $(X, Y) \in \mathcal{X} \times \mathcal{Y}$ denote a feature-label pair following a probability distribution $P$. Let $P^X, P^Y$ denote the marginal distributions of $X$ and $Y$. In this work, we consider the multi-class

classification problem where $\mathcal{Y} = \{0, \ldots K-1\}$, and assume $\mathcal{X} \subseteq \mathbb{R}^d$. Let $S = \{(\mathbf{x}_i, y_i)\}_{i=1}^n$ be $n$ i.i.d samples drawn from $P$. Let $P_n$ be the empirical distribution of $S$ and $P_n^X, P_n^Y$ be the marginal distributions of $\{\mathbf{x}_i\}_{i=1}^n, \{y_i\}_{i=1}^n$.

In classification, our goal is to find a predictor that can well predict the label of any feature from just the samples $S$. Let $f : \mathcal{X} \to \mathbb{R}^K$ denote a score-based classifier which assigns $X$ to class $\operatorname{argmax}_i f_i(X)$. The expected classification risk of $f$ is defined as $\mathbb{E}_{X,Y}\left[\ell_{0-1}(f(X), Y)\right]$, where $\ell_{0-1}(f(X), Y) = 0$ if $\operatorname{argmax}_i f_i(X) = Y$, and 1 otherwise. Since optimizing 0/1 risk is computationally intractable, we consider convex surrogates of $\ell_{0-1}(f(X), Y)$, which we denote by $\ell(f(X), Y)$; typical choices for $\ell$ include the logistic loss and the exponential loss. The population risk of $f$ is then defined as $R(f) = \mathbb{E}_{X,Y}\left[\ell(f(X), Y)\right]$. Since directly optimizing the population risk is impossible, we approximate it with the empirical risk $\widehat{R}_S(f) = \frac{1}{n}\sum_{i=1}^n \ell(f(\mathbf{x}_i), y_i)$ and try to find its minimizer.

We consider classifiers of the form $f(X) = W\phi(X)$, where $\phi : \mathcal{X} \to \mathbb{R}^D$ is the feature transformer and $W \in \mathbb{R}^{K \times D}$ is the linear classifier on top. A popular choice for $\phi$ is a neural network. We denote the population and empirical risks of such an $f$ as $R(W, \phi), \widehat{R}_S(W, \phi)$. We usually work in the space of feature transforms. Let $L_2(P)$ denote the space of square integrable functions w.r.t $P$, and define the inner product between $\phi_1, \phi_2 \in L_2(P)$ as $\langle \phi_1, \phi_2 \rangle_P = \mathbb{E}_{X \sim P}\left[\langle \phi_1(X), \phi_2(X) \rangle\right]$. We denote with $\nabla_\phi R(W, \phi)$ the functional gradient of $R(W, \phi)$ w.r.t $\phi$ in the $L_2(P^X)$ space, which is defined as $\nabla_\phi R(W, \phi)(\mathbf{x}) = \mathbb{E}_{Y|\mathbf{x}}\left[W^T \nabla \ell(W\phi(\mathbf{x}), Y)\right]$, where $\nabla \ell(W\phi(\mathbf{x}), y)$ denotes the gradient of $\ell$ w.r.t its first argument, evaluated at $W\phi(\mathbf{x})$. Similarly, we let $\nabla_\phi \widehat{R}_S(W, \phi)$ denote the functional gradient of $\widehat{R}_S(W, \phi)$ in the $L_2(P_n^X)$ space

$$\nabla_\phi \widehat{R}_S(W, \phi)(\mathbf{x}) = \begin{cases} W^T \nabla \ell(W\phi(\mathbf{x}_i), y_i), & \text{if } \mathbf{x} = \mathbf{x}_i, \\ 0 & \text{otherwise} \end{cases}.$$

**Additive Boosting.** In this work, we refer to traditional boosting as additive boosting, as it constructs additive models of weak classifiers. Let $\mathcal{F}$ be a hypothesis class of weak classifiers, a typical example being decision trees of bounded depth. Additive boosting aims to find an element in the linear span of $\mathcal{F}$ which minimizes the empirical risk $\widehat{R}_S(f)$. As previously mentioned, there exists a duality between boosting and greedy algorithms [18, 19, 26]. Many popular boosting algorithms use a greedy forward stagewise approach to find a minimizer of $\widehat{R}_S(f)$, and solve the following in each iteration:

$$\eta_t, f_t = \operatorname{argmin}_{\eta \in \mathbb{R}, f \in \mathcal{F}} \widehat{R}_S\left(\sum_{i=1}^{t-1} \eta_i f_i + \eta f\right),$$

where $\eta$ is the learning rate. Various algorithms differ in how they solve this optimization problem. In gradient boosting, one uses a linear approximation of $\widehat{R}_S$ around $\sum_{i=1}^{t-1} \eta_i f_i$ [26]. In this work, we take this greedy view of boosting to design the generalized boosting framework.

**Additive Representation Boosting.** In this work, we perform boosting in the representation space, contrasting with traditional boosting which works in the output space. Let $\mathcal{G}$ be a hypothesis class of *weak feature transformers*, whose examples include the set of one layer neural networks of bounded width and a set of vector-valued polynomials of bounded degree. More generally, $\mathcal{G}$ can be any set of non-linear transformations. In additive representation boosting, we aim to find a strong feature transform $\phi$ in the linear span of $\mathcal{G}$, and a linear predictor $W \in \mathcal{W} \subseteq \mathbb{R}^{K \times D}$ that minimizes $\widehat{R}_S(W, \phi)$. To this end, we consider greedy algorithms that solve the following problem each iteration:

$$W_t, g_t = \operatorname{argmin}_{W \in \mathcal{W}, g \in \mathcal{G}} \widehat{R}_S\left(W, \phi_{t-1} + \eta_t g\right), \tag{1}$$

where $\phi_t = \phi_0 + \sum_{i=1}^t \eta_i g_i$ with $\phi_0$ being the initial feature transformation, and $\{\eta_i\}_{i=1}^\infty$ is a predefined learning rate schedule.

## 3 Generalized Boosting

The starting point for our generalized boosting framework is the additive representation boosting described in Section 2. Typically, linear combinations of weak feature transformations are not powerful enough to model complex decision boundaries. Consequently, the minimizer of $\widehat{R}_S(W, \phi)$ over the linear span of $\mathcal{G}$ tends to have a high risk. A simple workaround for this issue would be to perform additive boosting with a complex hypothesis class $\mathcal{G}$. For example, if the weak feature transformers are one layer neural networks, then one could increase the complexity of $\mathcal{G}$ by using deeper networks. However, such an alternative has several drawbacks both from an optimization

and generalization perspective and defeats the purpose of boosting, which aims to convert weak learners into strong learners. From an optimization perspective, moving to complex $\mathcal{G}$ makes each greedy step harder to optimize. For example, compared to deep neural networks, shallow networks are easier to optimize, require fewer resources, and are easier to analyze or interpret [5]. From a generalization perspective, since the generalization bounds of boosting depend on the complexity of $\mathcal{G}$, larger hypothesis classes can lead to overfitting and poor performance on unseen data.

In this work, we are interested in other approaches for increasing the complexity of models produced by boosting, while ensuring the boosting/greedy steps are easy to implement. One way to achieve this is by considering more complex combinations of weak feature transformers than the linear combinations considered in additive representation boosting. Formally, let $\mathcal{G}_t$ denote the hypothesis class of feature transformations used in the $t^{th}$ iteration of boosting. In additive boosting, $\mathcal{G}_t = \mathcal{G}$ for all $t$. In our generalized boosting framework, we increase the complexity of $\mathcal{G}_t$ by letting it depend on the past iterates $\{\phi_i\}_{i=0}^{t-1}$. Here are some potential choices for $\mathcal{G}_t$, other than the ones stated in the introduction: $\{g \circ (\sum_{i=0}^{t-1} \alpha_i \phi_i), \text{ for } g \in \mathcal{G}, \alpha_i \in \mathbb{R}\}$, $\{g \circ \phi_{t-1} \circ \phi_{t-2} \cdots \circ \phi_0, \text{ for } g \in \mathcal{G}\}$. Depending on the problem domain, one could consider several other ways of constructing $\mathcal{G}_t$ using the past iterates. Note that even with these complex choices of $\mathcal{G}_t$, the greedy steps are easy to implement and only need a weak learner which can identify an element in $\mathcal{G}$ that best fits the data. As a result, this remains in the spirit of boosting and at the same time ensures the models learned are complex enough for real world problems.

We now present our algorithm for generalized boosting (see Algorithm 1). Similar to additive representation boosting, our algorithm proceeds in a greedy fashion. In the $t^{th}$ iteration of the algorithm, we aim to solve the following optimization problem:

$$W_t, g_t = \operatorname*{argmin}_{W \in \mathcal{W}, g \in \mathcal{G}_t} \widehat{R}_S \left( W, \phi_{t-1} + \eta_t g \right). \tag{2}$$

We provide two approaches for solving this problem. One is the *exact greedy approach*, which directly solves the optimization problem (Algorithm 2). For problems where direct optimization of Equation (2) is difficult[1], we provide an approximate technique which performs functional gradient descent on the objective. In this approach, which we call *gradient greedy approach*, we approximate the objective with the linear approximation of $\widehat{R}_S$ around $\phi_{t-1}$ (Algorithm 3):

$$\widehat{R}_S \left( W, \phi_{t-1} + \eta_t g \right) \approx \widehat{R}_S \left( W, \phi_{t-1} \right) + \eta_t \left\langle \nabla_\phi \widehat{R}_S(W, \phi_{t-1}), g \right\rangle_{P_n^X}.$$

To optimize the linear approximation, we first fix $W$ to $W_{t-1}$ and find a minimizing $g_t \in \mathcal{G}_t$. Intuitively, this step can be seen as finding a $g$ which best aligns with the negative functional gradient of empirical risk at the current iterate. For appropriate choice of learning rate $\eta$, moving along $g_t$ results in reduction of $\widehat{R}_S$. Next, we fix $g_t$ and find a linear predictor $W$ which minimizes the empirical risk $\widehat{R}_S(W, \phi_t)$. This alternating optimization of $g$ and $W$ makes the algorithm easy to implement in practice. Moreover, this algorithm is more stable than joint optimization of $g$ and $W$. We note that such gradient greedy approaches have been developed for traditional boosting [26].

### 3.1 Compositional Boosting

As one particular instantiation of our framework, we consider $\mathcal{G}_t$'s constructed by composing elements from a weak feature transformer class $\mathcal{G}$ with the past iterates $\{\phi_i\}_{i=0}^{t-1}$ and study the resulting boosting algorithms. We refer to such boosting algorithms as *compositional boosting* algorithms since the strong feature transformer is constructed from weak feature transformer via function composition. When $\mathcal{G}_t = \{g \circ \phi_{t-1} \text{ for } g \in \mathcal{G}\}$, the models in our framework have the ResNet architecture and can be defined recurrently as $\phi_t = \phi_{t-1} + \eta_t g_t \circ \phi_{t-1}$. Moreover, Algorithm 1 with this choice of $\mathcal{G}_t$ and Algorithm 2 as update routine give us the greedy layer-wise supervised training technique proposed by Bengio et al. [6] and recently revisited by Belilovsky et al. [5]. In another recent work, Huang et al. [22] propose a boosting-based algorithm for learning ResNets (see Algorithm 4 in Appendix). We now show that their approach is equivalent to the greedy technique of Bengio et al. [6], and thus can be seen as an instance of our general framework. We note that such a connection is not known previously.

**Proposition 3.1.** *Suppose the classification loss $\ell$ is the exponential loss. Then the greedy technique of Huang et al. [22] for learning ResNets is equivalent to the greedy layer-wise supervised training technique of Bengio et al. [6].*

In another recent work, Nitanda and Suzuki [29] propose a gradient boosting technique to greedily learn a ResNet. This algorithm is closely related to the gradient greedy approach described in Algorithm 3, with $\mathcal{G}_t = \{g \circ \phi_{t-1} \text{ for } g \in \mathcal{G}\}$.

---

**Algorithm 1** Generalized Boosting

1: **Input:** Training data $S = \{(\mathbf{x}_i, y_i)\}_{i=1}^n$, iterations $T$, initial linear predictor $W_0$, initial feature transformer $\phi_0$, learning rates $\{\eta_i\}_{i=1}^T$, Update-routine: UPDATE
2: $t \leftarrow 1$
3: **while** $t \leqslant T$ **do**
4:　　Construct feature transformer class $\mathcal{G}_t$ based on past iterates $\{(W_i, \phi_i)\}_{i=0}^{t-1}$
5:　　$W_t, \phi_t, g_t \leftarrow \text{UPDATE}\,(S, W_{t-1}, \phi_{t-1}, \eta_t, \mathcal{G}_t)$
6:　　$t \leftarrow t + 1$
7: **end while**
8: **Return:** $W_T, \phi_T$

---

| **Algorithm 2** Exact Greedy Update | **Algorithm 3** Gradient Greedy Update |
|---|---|
| 1: **Input:** Training data $S$, previous iterate $(W, \phi)$, learning rate $\eta$, feature transformer class $\mathcal{G}$<br>2:<br>$$W^+, g^+ \leftarrow \underset{\widetilde{W} \in \mathcal{W}, \tilde{g} \in \mathcal{G}}{\operatorname{argmin}} \widehat{R}_S(\widetilde{W}, \phi + \eta\tilde{g})$$<br>3: $\phi^+ \leftarrow \phi + \eta g^+$<br>4: **Return:** $W^+, \phi^+, g^+$ | 1: **Input:** Training data $S$, previous iterate $(W, \phi)$, learning rate $\eta$, feature transformer class $\mathcal{G}$<br>2: // Pick a descent direction<br>3: $g^+ \leftarrow \operatorname{argmin}_{\tilde{g} \in \mathcal{G}} \left\langle \nabla_\phi \widehat{R}_S(W, \phi), \tilde{g} \right\rangle_{P_n^X}$<br>4: $\phi^+ \leftarrow \phi + \eta g^+$<br>5: // Update the linear predictor<br>6: $W^+ \leftarrow \operatorname{argmin}_{\widetilde{W} \in \mathcal{W}} \widehat{R}_S(\widetilde{W}, \phi^+)$<br>7: **Return:** $W^+, \phi^+, g^+$ |

---

We now highlight certain drawbacks of the existing greedy layer-wise training techniques, which arise from the particular choice of $\mathcal{G}_t$ used by these algorithms. Since $\{g \circ \phi_{t-1} \text{ for } g \in \mathcal{G}\}$ is constructed solely based on the past iterate $\phi_{t-1}$, any mistake in $\phi_{t-1}$ is propagated to all the future iterates. As a result, these algorithms can not recover from their past mistakes. As an example, consider the following scenario where two points $\mathbf{x}_1, \mathbf{x}_2$ belonging to two different classes are placed close to each other in the feature space, after $1^{st}$ iteration of greedy; that is $\phi_1(\mathbf{x}_1) \approx \phi_1(\mathbf{x}_2)$. In such a scenario, the future iterates $\{\phi_t\}_{t=2}^\infty$ generated by existing greedy algorithms will always place $\mathbf{x}_1, \mathbf{x}_2$ close to each other in the representation space. As a result, the algorithm will always misclassify at least one of $\mathbf{x}_1, \mathbf{x}_2$. Another issue with existing greedy techniques is that they do not guarantee that the complexity of $\mathcal{G}_t$ increases with time $t$. In such scenarios, Algorithm 1 doesn't make much progress in each iteration and can result in poor models. As an example, consider the setting where $\mathcal{G}$ is the set of all linear transformations. Suppose $\phi_0$ is the identity transform and $\phi_1$ is such that its range lies in a low dimensional subspace. Then, it is evident that $\mathcal{G}_1 \supseteq \mathcal{G}_t$ for all $t \geqslant 2$.

To fix these issues, we propose two new compositional boosting algorithms obtained with a more careful choice of $\mathcal{G}_t$. In our first algorithm, which we call DenseCompBoost, we choose $\mathcal{G}_t$ as follows

$$\mathcal{G}_t = \left\{ g \circ \left( \text{Id} + \sum_{i=0}^{t-1} \alpha_i \phi_i \right), \text{ for } g \in \mathcal{G}, \alpha_i \in \mathbb{R} \right\}, \tag{3}$$

where $\text{Id}(\cdot)$ is the identify function. Such a choice of $\mathcal{G}_t$ helps us recover from the past mistakes. For example, if $\phi_1$ is a constant function, then the algorithm can still learn a good feature transformer by relying on the input $\mathbf{x}$ and the initial feature transform $\phi_0$. Moreover, our choice of $\mathcal{G}_t$ ensures its complexity grows with $t$ and satisfies: $\mathcal{G}_{t-1} \subseteq \mathcal{G}_t$, for all $t$. We call our algorithm DenseCompBoost, since the resulting model for this choice of $\mathcal{G}_t$ resembles a DenseNet [23], where each layer is allowed to be connected to all the previous layers. That being said, the models output by DenseCompBoost differ from DenseNet in how they aggregate the previous layers. DenseNet concatenates the features from previous layers, whereas DenseCompBoost adds the features. Our second algorithm, which we call CmplxCompBoost, tries to increase the complexity of $\mathcal{G}_t$ in each iteration as follows

$$\mathcal{G}_t = \left\{ g \circ \phi_{t-1}, \text{ for } g \in \widetilde{\mathcal{G}}_t \right\}, \tag{4}$$

where $\widetilde{\mathcal{G}}_t$ is a weak feature transformer class and satisfies $\widetilde{\mathcal{G}}_{t-1} \subset \widetilde{\mathcal{G}}_t$ for all $t$. In the case of one layer neural networks, such $\widetilde{\mathcal{G}}_t$'s can be constructed by increasing the layer width with $t$. We note that the $\widetilde{\mathcal{G}}_t$ in this algorithm is independent of the past iterates. By increasing the complexity of $\widetilde{\mathcal{G}}_t$ with $t$, we expect the complexity of $\mathcal{G}_t$ to increase and Algorithm 1 to make more progress in each iteration. While not immediately evident, we note that this technique can also fix the mistakes made by past iterates. For example, suppose $\phi_1$ is such that it places two points $\mathbf{x}_1, \mathbf{x}_2$ from different classes,

close to each other in the feature space. Then having a more complex $\widetilde{\mathcal{G}}_2$ can help recover from this mistake, as one can potentially find a $g \in \widetilde{\mathcal{G}}_2$ which can separate these two points. In Section 5, we present empirical evidence showing that our new boosting algorithms have superior performance over existing additive and compositional boosting algorithms. Further empirical evidence corroborating the issues we identified with existing layer-wise training techniques can be found in Appendix J.1.

# 4 Excess Risk Bounds

In this section, we provide excess risk bounds for the models' output by the generalized boosting framework. Our results depend on a *weak learning condition* on the hypothesis class $\mathcal{G}_t$ used in the $t^{th}$ iteration of Algorithm 1. This condition is a way to quantify the relative strength of $\mathcal{G}_t$ and roughly says that there always exists an element in $\mathcal{G}_t$ which has an acute angle with the negative functional gradient at the current iterate. Such a condition ensures progress in each iteration of boosting.

**Definition 4.1.** *Let $\beta \in (0, 1], \epsilon \geqslant 0$ be constants. $\mathcal{G}_{t+1}$ is said to satisfy the $(\beta, \epsilon)$-weak learning condition for a dataset $S$, if there exists a $g \in \mathcal{G}_{t+1}$ such that*

$$\left\langle g, -\nabla_\phi \widehat{R}_S(W_t, \phi_t) \right\rangle_{P_n^X} \geqslant \beta B(\mathcal{G}_{t+1}) \|\nabla_\phi \widehat{R}_S(W_t, \phi_t)\|_{P_n^X} - \epsilon,$$

*where $B(\mathcal{G}_{t+1}) = \sup_{g \in \mathcal{G}_{t+1}} \|g\|_{P_n^X}$, and $P_n$ is the empirical distribution of $S$.*

In traditional boosting, such conditions are typically referred to as the *edge* of a weak learner and play a crucial role in the convergence analysis. For example, Freund and Schapire [14] assume that for any set of weights over the training set $S$, there exists a classifier in the hypothesis class of weak classifiers which has better than random accuracy on the weighted samples. The following proposition shows that their condition is closely related to Definition 4.1.

**Proposition 4.1.** *For binary classification, the weak learning condition of Freund and Schapire [14] satisfies the empirical weak learning condition in Definition 4.1, albeit in the label space.*

For binary classification problems, it is well known that the weak learning condition of [14] is the weakest condition under which boosting is possible [15, 31]. This, together with the above proposition, suggests that our weak learning condition in Definition 4.1 cannot be weakened for binary classification problems.

To begin with, we derive excess risk bounds for the gradient greedy approach. Our analysis crucially relies on the observation that it can be viewed as performing inexact gradient descent on the population risk $R$. Several recent works have analyzed inexact gradient descent on convex objectives [2, 12, 33, 34]. However, the condition on the inexact gradient imposed by these works is different from ours and in many cases is stronger than our condition. For example, the condition of Balakrishnan et al. [2] translates to $\|g + \nabla_\phi R(W, \phi)\|_{P^X} \leqslant \epsilon$ in our setting, which is stronger than our weak learning condition. So the core of our analysis focuses on understanding inexact gradient descent with descent steps satisfying the weak learning condition in Definition 4.1. In our analysis, we consider a sample-splitting variant of the algorithm, where in each iteration we use a fresh batch of samples. This is mainly done to simplify the analysis by avoiding complex statistical dependencies between the iterates of the algorithm. Let $\tilde{n} = \lfloor \frac{n}{T} \rfloor$, we split the training dataset $S$ into $T$ subsets $\{S_t\}_{t=1}^T$ of size $\tilde{n}$, where $S_t = \{(\mathbf{x}_{t,i}, y_{t,i})\}_{i=1}^{\tilde{n}}$. We work with the subset $S_t$ in the $t^{th}$ iteration of Algorithm 1. We are now ready to state our main result on the excess risk bounds of the iterates of Algorithm 3. Our results depend on the Rademacher complexity terms related to the hypothesis sets $\mathcal{W}, \mathcal{G}_t$

$$\mathcal{R}(\mathcal{W}, \mathcal{G}_t) = \mathbb{E}\left[ \sup_{\substack{W \in \mathcal{W}, \\ g \in \mathcal{G}_t}} \frac{1}{\tilde{n}} \sum_{i=1}^{\tilde{n}} \sum_{k=1}^{K} \rho_{ik} [Wg(\mathbf{x}_{t,i})]_k \right], \ \mathcal{R}(\mathcal{G}_t) = \mathbb{E}\left[ \sup_{g \in \mathcal{G}_t} \frac{1}{\tilde{n}} \sum_{i=1}^{\tilde{n}} \sum_{j=1}^{D} \rho_{ij} [g(\mathbf{x}_{t,i})]_j \right],$$

where $[\mathbf{u}]_k$ denotes the $k^{th}$ entry of a vector $\mathbf{u}$, and the expectation is taken w.r.t the randomness from $S_t$ and the Rademacher random variables $\rho_{ij}$'s.

**Theorem 4.1** (Gradient Greedy). *Suppose the classification loss $\ell$ is $L$-Lipschitz and $M$-smooth w.r.t the first argument. Let the hypothesis set of linear predictors $\mathcal{W}$ be s.t. any $W \in \mathcal{W}$ satisfies $\lambda_{min}(WW^T) \geqslant \sigma_{min}^2 > 0$ and $\lambda_{max}(WW^T) \leqslant \sigma_{max}^2$. Moreover, suppose for all $t$, $\mathcal{G}_t$ satisfies the $(\beta, \epsilon_t)$-weak learning condition of Definition 4.1 for any dataset $S_t$. Finally, suppose any $g \in \mathcal{G}_t$ is bounded with $\sup_X \|g(X)\|_2 \leqslant B$. Let the learning rates $\{\eta_t\}_{t=1}^\infty$ be chosen as $\eta_t = ct^{-s}$, for some $s \in \left(\frac{\beta+1}{\beta+2}, 1\right)$ and positive constant $c$. If Algorithm 1 is run for $T$ iterations with Algorithm 3 as*

*update routine, then $(W_T, \phi_T)$, the $T^{th}$ iterate output by the algorithm, satisfies the following risk bound for any $W^*, \phi^*$ and $\alpha \in (0, \beta(1-s))$, with probability at least $1 - \delta$ over datasets of size $n$*

$$R(W_T, \phi_T) \leqslant R(W^*, \phi^*) + O\left(\frac{1}{T^\alpha} + T^{2-s}\sqrt{\frac{\log \frac{T}{\delta}}{\tilde{n}}}\right) + 2\sum_{t=1}^{T} \eta_t \left(L\mathcal{R}\left(\mathcal{W}, \mathcal{G}_t\right) + L\mathcal{R}\left(\mathcal{G}_t\right) + \epsilon_t\right).$$

***Proof Sketch.*** We first show that Algorithm 3 can be viewed as performing inexact gradient descent on the population risk $R$. Specifically, we show that with high probability, the $t^{th}$ iterate $g_t$ satisfies

$$\langle g_t, -\nabla_\phi R(W_{t-1}, \phi_{t-1})\rangle_P \geqslant \beta B \|\nabla_\phi R(W_{t-1}, \phi_{t-1})\|_P - \epsilon_t - \zeta_t,$$

for some $\zeta_t > 0$. This follows from the weak learning condition satisfied by $\mathcal{G}_t$. Ignoring $\epsilon_t, \zeta_t$, the above equation shows that $g_t$ makes acute angle with the population functional gradient at $\phi_{t-1}$. Consequently, we would expect the population risk to decrease, if we move along $g_t$. This is indeed the case, and the final step in the proof formalizes this intuition. $\qquad\square$

**Remarks:** We now briefly discuss the above result. See Appendix D for more discussion.
- The reference classifier $(W^*, \phi^*)$ in the above bound can be any classifier, as long as $\|W^*\|_2 < \infty, \|\phi^*\|_{PX} < \infty$. In particular, if there exists a Bayes optimal classifier satisfying this condition, then the above Theorem provides an excess risk bound w.r.t the Bayes optimal classifier.
- The $T^{-\alpha}$ term in the bound corresponds to the *optimization error*. The $\eta_t\epsilon_t$ term corresponds to the *approximation error* and the rest of the terms correspond to the *generalization error*. As $T$ increases, the optimization error goes down, and as $\tilde{n}$ increases, the generalization error goes down. If there is no approximation error, that is $\epsilon_t = 0$ for all $t$, then the excess risk goes down to $0$ as $\tilde{n}, T \to \infty$ at appropriate rate.
- If $\beta = 1$, then for appropriate choice of step size the optimization error goes down as $O\left(T^{-1/3+\gamma}\right)$, for some arbitrarily small $\gamma > 0$. This rate is slower than the $O(T^{-1})$ rates for inexact gradient descent obtained by Devolder et al. [12], Schmidt et al. [33]. However, we note that unlike our work, these works assume that the level sets of the objective are bounded. Under the assumption that the level sets of population risk are bounded, the optimization error in Theorem 4.1 can be improved to $O(T^{-1})$. However, such a condition need not hold in the our setting.
- Note that the risk bounds are modular and only depend on the Rademacher complexity terms $\mathcal{R}(\mathcal{W}, \mathcal{G}_t), \mathcal{R}(\mathcal{G}_t)$ which capture the complexity of $\mathcal{G}_t$. To instantiate Theorem 4.1 for specific choices of $\mathcal{G}_t$, we need to bound these two complexity terms.

We now extend the analysis of Theorem 4.1 to the exact greedy approach.

**Corollary 4.1** (Exact Greedy). *Consider the setting of Theorem 4.1. Suppose Algorithm 1 is run with Algorithm 2 as update routine. Then $(W_T, \phi_T)$, the $T^{th}$ iterate output by the algorithm, satisfies the same risk bounds as gradient greedy algorithm in Theorem 4.1.*

In the rest of the section, we instantiate Theorem 4.1 for specific choices of $\mathcal{G}_t$. We first consider the additive representation boosting algorithm.

**Corollary 4.2.** *Consider the setting of Theorem 4.1 and consider the additive representation boosting algorithm, where $\mathcal{G}_t = \mathcal{G}$ for all $t$. Suppose $\mathcal{G}$ is the set of one layer neural networks with sigmoid activation functions: $\mathcal{G} = \left\{\sigma(C\mathbf{x}), \text{ for } C \in \mathbb{R}^{D \times d}, \|C_{i,*}\|_1 \leqslant \Lambda, \forall i\right\}$. Moreover, suppose the feature domain $\mathcal{X}$ is a subset of $[0,1]^d$. Then the $T^{th}$ iterate output by Algorithm 1, with Algorithm 2 or 3 as update routine, satisfies the following risk bound for any $(W^*, \phi^*)$, with probability at least $1 - \delta$*

$$R(W_T, \phi_T) \leqslant R(W^*, \phi^*) + O\left(\frac{1}{T^\alpha}\right) + 2\sum_{t=1}^{T} \eta_t\epsilon_t + O\left(\frac{KD\Lambda T^{1-s}\log D}{\sqrt{\tilde{n}}} + T^{2-s}\sqrt{\frac{\log \frac{T}{\delta}}{\tilde{n}}}\right).$$

Next, we consider the layer-by-layer fitting technique of Bengio et al. [6].

**Corollary 4.3.** *Consider the setting of Corollary 4.2 and consider the layer-by-layer training technique of Bengio et al. [6], where $\mathcal{G}_t = \{g \circ \phi_{t-1} \text{ for } g \in \mathcal{G}\}$. Suppose $\mathcal{G}$ is the set of one layer neural networks with sigmoid activation functions: $\mathcal{G} = \left\{\sigma(C\mathbf{x}), \text{ for } C \in \mathbb{R}^{D \times D}, \|C_{i,*}\|_1 \leqslant \Lambda, \forall i\right\}$. Then the $T^{th}$ iterate output by Algorithm 1, with Algorithm 2 or 3 as update routine, satisfies the following risk bound for any $(W^*, \phi^*)$ with probability at least $1 - \delta$*

$$R(W_T, \phi_T) \leqslant R(W^*, \phi^*) + O\left(\frac{1}{T^\alpha}\right) + 2\sum_{t=1}^{T} \eta_t\epsilon_t + O\left(\frac{KD\Lambda T^{2-2s}\log D}{\sqrt{\tilde{n}}} + T^{2-s}\sqrt{\frac{\log \frac{T}{\delta}}{\tilde{n}}}\right).$$

Note that the generalization and optimization errors for both additive feature boosting and layer-by-layer fitting have similar dependence on $T, \tilde{n}$. However, the latter tends to have a smaller approximation error ($\epsilon_t$) as it is able to build complex $\mathcal{G}_t$'s over time. So one would expect layer-by-layer fitting to output models with a better population risk, which our empirical results in fact verify.

## 5    Experiments

In this section, we present experiments comparing the performance of various boosting techniques on both simulated and benchmark datasets.

**Baselines.** We compare our proposed boosting techniques with XGBoost, AdaBoost, additive representation boosting (discussed in Corollary 4.2) and greedy layer-by-layer training technique of Bengio et al. [6] (Corollary 4.3). XGBoost uses decision trees as weak classifiers. For AdaBoost, we use 1 hidden layer neural networks as weak classifiers. We use two kinds of neural networks, based on the dataset. For tabular datasets, we use fully connected networks and for image datasets, we use convolutional networks (CNN) with the convolution block made up of *Convolution, BatchNorm, ReLU* layers arranged sequentially. For additive representation boosting (Additive Feature Boost from now on) and layer-by-layer fitting (StdCompBoost from now on), the weak feature transformer class $\mathcal{G}$ consists of one layer neural network transformations. Similar to AdaBoost, we use two kinds of transformations: a) fully connected transformations of the form $g(\mathbf{x}) = \text{ReLU}(C\mathbf{x} + \mathbf{d})$, and b) convolutional transformations with *Convolution, BatchNorm, ReLU* blocks arranged sequentially. Finally, we also compare against end-to-end training of ResNets.

**Proposed Techniques.** For DenseCompBoost, we use a slight variant of $\mathcal{G}_t$ defined in Equation (3) : $\mathcal{G}_t = \{h + g \circ (\sum_{i=0}^{t-1} \alpha_i \phi_i), \text{ for } h \in \mathcal{H}, g \in \mathcal{G}, \alpha_i \in \mathbb{R}\}$, where $\mathcal{H}, \mathcal{G}$ are weak feature transformer classes. We use this variant because the dimensions of the input feature space and the representation space need not be the same, and as a consequence $\mathcal{G}_t$ in Equation (3) can not always be used. Similar to StdCompBoost, we consider two choices for $\mathcal{H}, \mathcal{G}$: one based on fully connected blocks and the other based on convolution blocks. For CmplxCompBoost, we again consider two choices for the weak transformer class $\tilde{\mathcal{G}}_t$ in Equation (4): a) $\text{ReLU}(C\mathbf{x} + \mathbf{d})$ with $C \in \mathbb{R}^{D_t \times D_{t-1}}$, where $D_t = D_{t-1} + \Delta$ for some positive constant $\Delta$, and b) convolution blocks with number of output channels equal to the number of input channels plus a constant $\Delta$. This choice of feature transformers ensures the complexity of $\tilde{\mathcal{G}}_t$ increases with $t$. We use exact greedy updates (Algorithm 2) for both of our proposed methods and set learning rate $\eta_t$ to 1. We do not present experimental results for Algorithm 3, which we noticed has marginally worse performance than Algorithm 2.

### 5.1    Simulated Datasets

**Datasets.** In this section we compare the techniques described above on simulated datasets. We generated 3 synthetic binary classification datasets in $\mathbb{R}^{32}$. Simulation 1 is a concentric ellipsoids dataset, where a point $\mathbf{x}$ is classified based on $\mathbf{x}^T A\mathbf{x}$, for some randomly generated positive semidefinite matrix $A$. Simulations 2, 3 are datasets whose classification boundaries are polynomials of degrees 8 and 9 respectively. For each of these datasets, we generated $10^6$ samples for training and testing.

**Hyper-parameters.** We used hold-out set validation to pick the best hyper-parameters for all the methods. We used 20% of the training data as validation data and picked the best parameters using grid search, based on validation accuracy. After picking the best parameters, we train on the entire training data and report performance on the test data. For all the greedy techniques based on neural networks, we used fully connected blocks and tuned the following parameters: weight decay, width of weak feature transformers, number of boosting iterations $T$, which we upper bound by 15. For CmplxCompBoost, we set $\Delta = D_0/5$. For end-to-end training, we tuned weight decay, width of layers, depth. We used SGD for optimization of all these techniques. The number of epochs and step size schedule of SGD are chosen to ensure convergence. For XGBoost, we tuned the number of trees, depth of each tree, learning rate. The exact values of hyper-parameters tuned for each of the methods can be found in Appendix J.

**Results.** Table 1 presents the results from our experiments. Both CmplxCompBoost and StdComp-Boost largely outperform the additive boosting methods, with CmplxCompBoost being slightly better due to the increasing complexity in $\tilde{G}_t$. Notably, DenseCompBoost performs significantly better than the rest and is able to bridge the gap between StdCompBoost and End-to-End. We attribute its success to its ability to recover from earlier mistakes: while StdCompBoost or CmplxCompBoost necessarily accumulate errors at each layer, DenseCompBoost is further connected to earlier layers, allowing it to undo its past mistakes.

Table 1: Test accuracy of various boosting techniques on synthetic datasets. Numbers in bold indicate the best performance among various greedy techniques.

| Technique | Simulation 1 | Simulation 2 | Simulation 3 |
|---|---|---|---|
| XGBoost (Trees) | 84.40 | 97.59 | 50.10 |
| AdaBoost (1 NN) | 67.90 | 93.73 | 72.64 |
| Additive Feature Boost | 88.49 | 93.91 | 73.13 |
| StdCompBoost | 91.53 | 96.95 | 82.49 |
| DenseCompBoost | **93.55** | **98.35** | **95.70** |
| CmplxCompBoost | 91.97 | 97.22 | 82.52 |
| End-to-End | 93.88 | 98.35 | 99.09 |

Table 2: Test accuracy of various boosting techniques on benchmark datasets. We use convolution blocks for the first 5 datasets and fully connected blocks for the other datasets.

| Technique | SVHN | FashionMNIST | CIFAR10 | Convex | MNIST-rot-back-image | MNIST | Letter | CovType | Connect4 |
|---|---|---|---|---|---|---|---|---|---|
| XGBoost (Trees) | 77.72 | 90.34 | 58.34 | 82.29 | 53.89 | 97.96 | 96.16 | **97.46** | 86.63 |
| AdaBoost (1 NN) | 82.88 | 88 | 72.78 | 86.17 | 50.02 | 98.27 | 92.08 | 90.95 | 86.39 |
| Additive Feature Boost | 83.36 | 89.95 | 74.33 | 89.30 | 54.31 | 98.27 | 90.86 | 93.12 | 86.58 |
| StdCompBoost | 90.81 | 92.77 | 81.93 | 98.19 | 73.17 | 98.37 | 96.43 | 95.61 | 86.33 |
| DenseCompBoost | 91.03 | **93.17** | 82.31 | **98.6** | 73.1 | 98.34 | **96.96** | 96.28 | **86.85** |
| CmplxCompBoost | **91.25** | **93.18** | **82.43** | 98.52 | **74.32** | 98.34 | 96.66 | 95.92 | 86.49 |
| End-to-End | 94.82 | 93.49 | 86.88 | 98.81 | 82.69 | 98.95 | 97.67 | 96.86 | 87.37 |

## 5.2 Benchmark Datasets

**Datasets.** In this section, we compare various techniques on the following image datasets: CIFAR10, MNIST, FashionMNIST [35], MNIST-rot-back-image [24], convex [35], SVHN [28], and the following tabular datasets from UCI repository [7]: letter recognition [17], forest cover type (covtype), connect4. The convex dataset involves classifying shapes in images as either convex or non-convex.

**Hyper-parameters.** For covtype dataset, which doesn't come with a test set, we randomly sample 20% of the original data and use it as the test set. We use a similar hyper-parameter selection technique as above and tune the same set of hyper-parameters as described above. We use convolution blocks for CIFAR10, SVHN, FashionMNIST, convex, MNIST-rot-back-image and fully connected blocks for the rest. We limit the width of fully connected blocks to 4096, and the number of output channels in convolution blocks to 128 while tuning the hyper-parameters for the composition boosting techniques and end-to-end training. For AdaBoost and additive representation boosting, we set these limits to 16000 and 350 respectively. For CmplxCompBoost with convolution blocks, we set $\Delta = D_0/8$. We *do not* use data augmentation in our experiments.

**Results.** Table 2 presents the results from our experiments. It can be seen that on image classification tasks, additive boosting techniques have poor performance. Among compositional boosting methods, StdCompBoost performs the worst. While DenseCompBoost performs comparably to CmplxCompBoost on image datasets, it is better on tabular data. We believe a hybrid of DenseCompBoost and CmplxCompBoost algorithms can achieve better performance than either of the algorithms.

## 6 Conclusion

We proposed a generalized framework for boosting, which allows for more complex forms of aggregation of weak learners than traditional boosting. Our generalized framework allows to derive learning algorithms that (a) have performance close to that of end-to-end trained DNNs, and (b) come with strong theoretical guarantees. Additive boosting algorithms do not satisfy property (a), while DNNs do not satisfy property (b). In particular, additive boosting algorithms, even with small neural networks as their weak classifiers, do not have the strong performance of end-to-end trained DNNs. Improving their performance requires the hypothesis space to increase in complexity while not increasing sample complexity of each boosting step too greatly, which can be achieved by our generalized boosting framework. One particular instantiation of our framework is aggregation using function compositions. A number of existing greedy techniques for learning neural networks fall into our framework, and our analysis allowed us to delineate some of their key flaws, then consequently, propose new techniques which improve upon them. We believe our work opens up a new line of inquiry for greedy learning of highly flexible models with rigorous theoretical guarantees, by leveraging the theory of boosting and generalized greedy algorithms in function spaces. We moreover believe our work has the potential to bridge the gap in performance between existing greedy layer-by-layer training techniques and end-to-end training of deep networks.

## Broader Impact

Deep learning has been tremendously successful over the past decade in many application areas such as computer vision, image recognition, speech recognition, and natural language processing. Despite this success, deep neural networks have largely remained a mystery. With millions of parameters, these models are blackboxes to humans, making it harder to diagnose errors. This also makes it harder to adopt these models in critical applications such as healthcare, law and finance. Consequently, it is crucial to come up with techniques that make neural networks transparent and easy to understand. We take a step towards this goal by drawing inspiration from classical boosting. Similar to classical boosting, our generalized boosting framework builds complex models greedily. But unlike classical boosting, it allows us to derive learning algorithms that have performance close to that of end-to-end trained DNNs. Moreover, models built using our framework are easy to understand and come with strong theoretical guarantees.

## Acknowledgement

We acknowledge the support of NSF via IIS-1909816, OAC-1934584, ONR via N000141812861, and Amazon Web Services (AWS).

## Footnotes

[1]Such scenarios can potentially arise if the feature transformations are non-differentiable functions.

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
