[Supplementary Material]

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

# A  Notation and Terminology

**Notation**

| Symbol | Description |
|---|---|
| $X$ | feature vector |
| $Y$ | label |
| $\mathcal{X}$ | domain of feature vector |
| $\mathcal{Y}$ | domain of the label |
| $K$ | number of classes in multi-class classification problem |
| $S$ | data set |
| $P$ | true data distribution |
| $P^X, P^Y$ | marginal distributions of $X, Y$ |
| $P_n$ | empirical distribution |
| $P_n^X, P_n^Y$ | empirical marginal distributions of $X, Y$ in data set $S$ |
| $f : \mathcal{X} \to \mathbb{R}^K$ | score based classifier |
| $\phi$ | feature transformer |
| $W$ | linear classifier on top of feature transformer |
| $\ell_{0-1}$ | 0/1 classification loss |
| $\ell$ | convex surrogate of $\ell_{0-1}$ |
| $R(f)$ | population risk of classifier $f$, measured w.r.t $\ell$ |
| $\widehat{R}_S(f)$ | empirical risk of classifier $f$, measured w.r.t $\ell$ |
| $R(W, \phi)$ | population risk of classifier $f = W\phi$, measured w.r.t $\ell$ |
| $\widehat{R}_S(W, \phi)$ | empirical risk of classifier $f = W\phi$, measured w.r.t $\ell$ |
| $L_2(P)$ | set of square integrable functions w.r.t $P$ |
| $f \circ g(\mathbf{x})$ | denotes function composition $f(g(\mathbf{x}))$ |
| $[\phi_0, \dots, \phi_t](\mathbf{x})$ | denotes concatenation of vectors $\phi_0(\mathbf{x}) \dots \phi_t(\mathbf{x})$ |
| $\mathcal{F}$ | hypothesis class of weak classifiers |
| $\mathcal{G}$ | hypothesis class of weak feature transformers |
| $\mathcal{G}_t$ | hypothesis class of weak feature transformers used in the $t^{th}$ iteration of greedy |
| $\mathcal{W}$ | hypothesis class of linear classifiers on top of feature transformers |

**Terminology**

| Term | Description |
|---|---|
| *Additive Boosting* | Classical boosting framework which constructs a strong classifier using additive combinations of weak classifiers |
| *Additive Feature Boosting* | Feature boosting framework which constructs a strong classifier using additive combinations of weak feature transformers with a linear classifier on top of the feature transformer |
| *Weak classifier* | Any classifier which by itself doesn't achieve good performance on a given classification task and whose performance we wish to boost |
| *Weak feature transformer* | Any feature transformation which by itself doesn't provide good performance on a given classification task and whose performance we wish to boost |

# B  Proof of Proposition 3.1

**Notation.**  We use the notation of  Huang et al. [22] in this proof. We note that this notation will only be used in this section. Later sections use the notation introduced in Section 2. We let $g_t(\mathbf{x})$ be the output of the $t^{th}$ residual block, which is given by the following recursion

$$g_t(\mathbf{x}) = f_{t-1} \circ g_{t-1}(\mathbf{x}) + g_{t-1}(\mathbf{x}) = \sum_{i=0}^{t-1} f_i \circ g_i(\mathbf{x}),$$

with $g_0, f_0$ equal to identity functions. The final output of a depth-$T$ ResNet, given input $\mathbf{x}$, is rendered after a linear classifier $W \in \mathbb{R}^{K \times D}$ on representation $g_{T+1}(\mathbf{x})$. Let $W_t$ be the auxiliary

linear classifier on top of the residual block $g_t$. Define $o_t(\mathbf{x})$ as

$$o_t(\mathbf{x}) \stackrel{def}{=} W_t g_t(\mathbf{x}).$$

Note that $o_t(\mathbf{x}) = \sum_{i=0}^t W_t f_i \circ g_i(\mathbf{x})$. Define $h_t(\mathbf{x})$ as $h_t(\mathbf{x}) \stackrel{def}{=} \alpha_{t+1} o_{t+1}(\mathbf{x}) - \alpha_t o_t(\mathbf{x})$, where $\alpha_t$ is a scalar. Huang et al. [22] consider exponential loss in their work, which is defined as

$$\ell(o(\mathbf{x}), y) = \sum_{k \neq y} \exp\left([o(\mathbf{x})]_k - [o(\mathbf{x})]_y\right).$$

**Algorithm of Bengio et al. [6].**   Using this notation, the greedy layer-by-layer training technique of Bengio et al. [6] for learning ResNets is given by the following update rule

$$W_{t+1}, f_t \leftarrow \operatorname*{argmin}_{W,f} \frac{1}{n} \sum_{i=1}^n \ell\left(W\left[f \circ g_t(\mathbf{x}_i) + g_t(\mathbf{x}_i)\right], y_i\right). \tag{5}$$

**Algorithm of Huang et al. [22].**   The algorithm of Huang et al. [22] for greedy learning of ResNets is given in Algorithm 4, which is a reproduction of Algorithm 3 of Huang et al. [22]. Note that the key update step is given in step 2 of Algorithm 5

$$f_t, \alpha_{t+1}, W_{t+1} \leftarrow \operatorname*{argmin}_{f,\alpha,W} \sum_{i=1}^n \ell(\alpha W[f \circ g_t(\mathbf{x}_i) + g_t(\mathbf{x}_i)], y_i). \tag{6}$$

Since $\alpha$ is a scalar, it can be consumed into the linear classifier $W$. This shows that the update step of Huang et al. [22] is equivalent to Equation (5).

---

**Algorithm 4** Greedy algorithm of Huang et al. [22] for learning ResNets

1: **Input:** Training data $S = \{(\mathbf{x}_i, y_i)\}_{i=1}^n$, iterations $T$, threshold $\gamma$
2: Initialize $t \leftarrow 0, \tilde{\gamma}_0 \leftarrow 0, \alpha_0 \leftarrow 0, o_0 \leftarrow \mathbf{0} \in \mathbb{R}^K, s_0(\mathbf{x}_i) = \mathbf{0} \in \mathbb{R}^K, \forall i \in [n]$
3: Initialize cost function $[C_0(i)]_k \leftarrow \begin{cases} 1 & \text{if } k \neq y_i \\ 1 - K & \text{if } k = y_i \end{cases}, \forall i \in [n], k \in [K]$
4: **while** $\gamma_t > \gamma$ **do**
5:     $f_t, \alpha_{t+1}, W_{t+1}, o_{t+1} \leftarrow$ Algorithm 5$(g_t)$
6:     Compute $\gamma_t \leftarrow \sqrt{\frac{\tilde{\gamma}_{t+1}^2 - \tilde{\gamma}_t^2}{1 - \tilde{\gamma}_t^2}}$, where $\tilde{\gamma}_{t+1} = \frac{-\sum_{i=1}^n C_t(i)^T o_{t+1}(\mathbf{x}_i)}{\sum_{i=1}^n \sum_{k \neq y_i} [C_t(i)]_k}$
7:     Update $s_{t+1}(\mathbf{x}_i) \leftarrow s_t(\mathbf{x}_i) + h_t(\mathbf{x}_i)$, where $h_t(\mathbf{x}_i) = \alpha_{t+1} o_{t+1}(\mathbf{x}_i) - \alpha_t o_t(\mathbf{x}_i)$
8:     Update cost function $[C_{t+1}(i)]_k \leftarrow \begin{cases} \exp\left([s_{t+1}(\mathbf{x}_i)]_k - [s_{t+1}(\mathbf{x}_i)]_{y_i}\right) & \text{if } k \neq y_i \\ -\sum_{k' \neq y_i} \exp\left([s_{t+1}(\mathbf{x}_i)]_{k'} - [s_{t+1}(\mathbf{x}_i)]_{y_i}\right) & \text{if } k = y_i \end{cases}, \forall i \in$
    $[n], k \in [K]$
9:     $t \leftarrow t + 1$
10: **end while**
11: $T \leftarrow t - 1$
12: **Return:** $W_{T+1}, \{f_t(\cdot), \forall t\}$

---

**Algorithm 5** Training a ResNet module

1: **Input:** $g_t$
2: $(f_t, \alpha_{t+1}, W_{t+1}) \leftarrow \operatorname{argmin}_{f,\alpha,W} \sum_{i=1}^n \ell(\alpha W[f \circ g_t(\mathbf{x}_i) + g_t(\mathbf{x}_i)], y_i)$
3: $o_{t+1}(\mathbf{x}) = W_{t+1}[f_t \circ g_t(\mathbf{x}) + g_t(\mathbf{x})]$
4: **Return:** $f_t, \alpha_{t+1}, W_{t+1}, o_{t+1}$

---

## C   Proof of Proposition 4.1

Freund and Schapire [14] consider the problem of binary classification with $\mathcal{Y} = \{-1, +1\}$. Let $\mathcal{F}$ be a hypothesis space of weak classifiers mapping $\mathcal{X}$ to $\mathcal{Y}$. Freund and Schapire [14] consider the following weak learning condition. For any set of non-negative weights $\{w_i\}_{i=1}^n$ over points

$\{(\mathbf{x}_i, y_i)\}_{i=1}^n$ such that $\sum_i w_i = 1$, there is a classifier $f \in \mathcal{F}$ which achieves an error at most $\frac{1}{2} - \frac{\beta}{2}$, for some $\beta > 0$. That is, there exists $f \in \mathcal{F}$ such that

$$\sum_{i=1}^n w_i \mathbb{I}(y_i \neq f(\mathbf{x}_i)) \leqslant \frac{1}{2} - \frac{\beta}{2}.$$

This can equivalently be written as

$$
\begin{aligned}
\sum_{i=1}^n w_i y_i f(\mathbf{x}_i) &= \sum_{i:y_i = f(\mathbf{x}_i)} w_i y_i f(\mathbf{x}_i) - \sum_{i:y_i \neq f(\mathbf{x}_i)} w_i y_i f(\mathbf{x}_i) + 2\sum_{i:y_i \neq f(\mathbf{x}_i)} w_i y_i f(\mathbf{x}_i) \\
&= 1 + 2\sum_{i:y_i \neq f(\mathbf{x}_i)} w_i y_i f(\mathbf{x}_i) \\
&\geqslant \beta \\
&= \beta \left(\sum_{i=1}^n w_i\right)
\end{aligned}
$$

(7)

We now show that this condition implies Definition 4.1 in the label space. We first introduce the notion of inner product between functions mapping $\mathcal{X}$ to $\mathbb{R}$. For any $f, g$ mapping $\mathcal{X}$ to $\mathbb{R}$, we define $\langle f, g\rangle_n$ as

$$\langle f, g\rangle_n = \frac{1}{n}\sum_{i=1}^n f(\mathbf{x}_i)g(\mathbf{x}_i).$$

Let the classification loss $\ell$ be such that $\ell(f(\mathbf{x}), y) = c(yf(\mathbf{x}))$ for some decreasing function $c : \mathbb{R} \to \mathbb{R}$. All the popular classification losses such as logistic, exponential, hinge losses satisfy this assumption. The functional gradient of $\widehat{R}_S$ w.r.t $f$ in the above inner product space is defined as

$$\nabla_f \widehat{R}_S(f)(\mathbf{x}) = \begin{cases} y_i c'(y_i f(\mathbf{x}_i)), & \text{if } \mathbf{x} = \mathbf{x}_i \\ 0, & \text{otherwise} \end{cases},$$

where $c'(z)$ is the derivative of $c$ at $z$. Note that since $c$ is a decreasing function, $c'(z) < 0$ for any $z$. Using this notation, it is easy to see that any hypothesis class $\mathcal{F}$ satisfying Equation (7) satisfies the following condition for any function $h : \mathcal{X} \to \mathbb{R}$

$$\exists f \in \mathcal{F}, \quad \left\langle f, -\nabla_f \widehat{R}_S(h)\right\rangle_n \geqslant \beta\|\nabla_f \widehat{R}_S(h)\|_1 \geqslant \frac{\beta}{\sqrt{n}}\|\nabla_f \widehat{R}_S(h)\|_n,$$

where $\|\nabla_f \widehat{R}_S(h)\|_1 = n^{-1}\sum_{i=1}^n |\nabla_f \widehat{R}_S(h)(\mathbf{x}_i)|$. This can be shown by substituting $w_i$ in Equation (7) with $-c'(y_i h(\mathbf{x}_i))$. This shows that the weak learning condition of Freund and Schapire [14] satisfies the weak learning condition in Definition 4.1, albeit in the label space.

## D  Discussion of Theorem 4.1

In this section, we discuss the results of Theorem 4.1.

**Remark D.1** (Reference Classifier). *The reference classifier $(W^*, \phi^*)$ in the bound in Theorem 4.1 can be any classifier, as long as $\|W^*\|_2 < \infty, \|\phi^*\|_{P^X} < \infty$. In particular, if there exists a Bayes optimal classifier satisfying this condition, then the above Theorem provides an excess risk bound w.r.t the Bayes optimal classifier.*

**Remark D.2** (Breakdown of Rates). *The $T^{-\alpha}$ term in the bound corresponds to the* optimization error. *The $\eta_t \epsilon_t$ term corresponds to the* approximation error *and the rest of the terms correspond to the* generalization error. *As $T$ increases, the optimization error goes down, and as $\tilde{n}$ increases, the generalization error goes down. If there is no approximation error, that is $\epsilon_t = 0$ for all $t$, then the excess risk goes down to 0 as $\tilde{n}, T \to \infty$ at appropriate rate.*

**Remark D.3** (Optimization Error). *If $\beta = 1$, then for appropriate choice of step size the optimization error goes down as $O\left(T^{-1/3+\gamma}\right)$, for some arbitrarily small $\gamma > 0$. This rate is slower than the $O(T^{-1})$ rates for inexact gradient descent obtained by Devolder et al. [12], Schmidt et al. [33]. However, we note that unlike our work, these works assume that the level sets of the objective are bounded. Under the assumption that the level sets of population risk are bounded, the optimization error in Theorem 4.1 can be improved to $O(T^{-1})$. However, such a condition need not hold in the our setting.*

**Remark D.4** (Lipschitzness of loss). *The assumptions of smoothness and Lipschitzness on $\ell$ are satisfied by popular loss functions such as logistic loss,* softmax + cross entropy loss. *Consider logistic loss for binary classification $\ell(z, y) = \log(1 + e^{-yz})$. It is easy to verify that $\ell(z, y)$ is 1-Lipschitz and 1-smooth w.r.t. $z$. Similarly, the softmax + cross entropy loss, which is given by, $\ell(\mathbf{z}, y) = -\mathbf{z}[y] + \log\left(\sum_{k=1}^{K} e^{\mathbf{z}[k]}\right)$ is 1-Lipschitz and 1-smooth w.r.t. $\mathbf{z}$.*

**Remark D.5** (Bounded Feature Transformers). *The boundedness assumption on the functions in $\mathcal{G}_t$ is satisfied by neural networks made up of bounded activation functions such as* sigmoid, tanh.

**Remark D.6** (Modular Bounds). *Note that the risk bounds are modular and only depend on the Rademacher complexity terms $\mathcal{R}(\mathcal{W}, \mathcal{G}_t), \mathcal{R}(\mathcal{G}_t)$ which capture the complexity of $\tilde{\mathcal{G}}_t$. To instantiate Theorem 4.1 for specific choices of $\mathcal{G}_t$, we need to bound these two complexity terms.*

**Remark D.7** (Bounds on 0/1 risk). *Since 0/1 loss is upper bounded by surrogate losses such as exponential, logistic loss, our Theorem also provides generalization bounds for 0/1 loss.*

**Remark D.8** (Sample Splitting). *A natural question that might arise regarding sample splitting is: "does this make our approach similar to bagging and random forests (RFs)?". We would like to note that even with sample splitting, our approach is not similar to bagging and RFs. Bagging and RFs create ensembles by independently training each base learner. Whereas, in boosting, the base learners are fit greedily and are not independent of each other. Another important distinction between RFs and boosting is that RFs work with complex base classifiers with good predictive power and aim to reduce the variance of these classifiers by averaging the predictions of multiple independently trained base classifiers. Whereas in boosting, one works with base classifiers with very little predictive power (i.e., high bias) and combines multiple such base classifiers to create a strong classifier with good predictive power (i.e., low bias). Viewed this way, our approach is very similar to boosting than RFs.*

# E Proof of Theorem 4.1

## E.1 Intermediate Results

In this section we present some intermediate results which we use in the proof of Theorem 4.1. The proof of the Theorem can be found in Section E.2.

**Lemma E.1.** *Consider the setting of Theorem 4.1. Let $(W_t, \phi_t)$ be the $t^{th}$ iterate generated by Algorithm 1 with Algorithm 3 as update routine. Then for any $t$, the following holds with probability at least $1 - \delta$ over datasets of size $n$*

$$R(W_t, \phi_t) \leqslant R(W_{t-1}, \phi_t) + 2\eta_t L\mathcal{R}(\mathcal{W}, \mathcal{G}_t) + \frac{4c\sigma_{max}BLt^{1-s}}{1-s}\left(\sqrt{\frac{\log 2/\delta}{\tilde{n}}} + \sqrt{\frac{K}{\tilde{n}}}\right),$$

*where $\mathcal{R}(\mathcal{W}, \mathcal{G}_t)$ is the Rademacher complexity term, which is defined as*

$$\mathcal{R}(\mathcal{W}, \mathcal{G}_t) = \mathbb{E}\left[\sup_{\substack{W \in \mathcal{W}, \\ g \in \mathcal{G}_t}} \frac{1}{\tilde{n}}\sum_{i=1}^{\tilde{n}}\sum_{k=1}^{K}\rho_{ik}[Wg(\mathbf{x}_{t,i})]_k\right],$$

*and the expectation is over the randomness from $S_t, \rho$'s.*

*Proof.* Throughout the proof, we condition on the past datasets $S_1, \ldots S_{t-1}$ and show that the Lemma holds for any choice of $S_1, \ldots S_{t-1}$. Consider the following upper bound for $R(W_t, \phi_t)$

$$R(W_t, \phi_t) \leqslant \widehat{R}_{S_t}(W_t, \phi_t) + \sup_{W \in \mathcal{W}, g \in \mathcal{G}_t}|R(W, \phi_{t-1} + \eta_t g) - \widehat{R}_{S_t}(W, \phi_{t-1} + \eta_t g)|$$

$$\overset{(a)}{\leqslant} \widehat{R}_{S_t}(W_{t-1}, \phi_t) + \sup_{W \in \mathcal{W}, g \in \mathcal{G}_t}|R(W, \phi_{t-1} + \eta_t g) - \widehat{R}_{S_t}(W, \phi_{t-1} + \eta_t g)|$$

$$\leqslant R(W_{t-1}, \phi_t) + 2\sup_{W \in \mathcal{W}, g \in \mathcal{G}_t}|R(W, \phi_{t-1} + \eta_t g) - \widehat{R}_{S_t}(W, \phi_{t-1} + \eta_t g)|,$$

where $(a)$ follows from the definition of $W_t$. We now rely on Rademacher complexity bounds in Theorem I.2 to bound the supremum in the RHS. To apply the bound, we first need to ensure

$\ell(W\phi_{t-1}(\mathbf{x}) + \eta_t Wg(\mathbf{x}), y)$ is bounded. Since $\sup_X \|g(X)\|_2 \leqslant B$ and $\lambda_{\max}\left(WW^T\right) \leqslant \sigma_{\max}^2$, it is easy to see that

$$\sup_X \|W\phi_{t-1}(\mathbf{x}) + \eta_t Wg(\mathbf{x})\|_2 \leqslant \sigma_{\max}B \sum_{i=1}^{t} \eta_i \leqslant \frac{c\sigma_{\max}Bt^{1-s}}{1-s},$$

where the last inequality follows from the definition of $\eta_t$. Since $\ell$ is $L$-Lipschitz in its first argument, we can show that $\ell(W\phi_{t-1}(\mathbf{x}) + \eta_t Wg(\mathbf{x}), y)$ lies in an interval of width $\frac{2c\sigma_{\max}BLt^{1-s}}{1-s}$. Applying Theorem I.2, we get with probability at least $1-\delta$

$$R(W_t, \phi_t) \leqslant R(W_{t-1}, \phi_t) + 2\mathbb{E}\left[\sup_{W\in\mathcal{W}, g\in\mathcal{G}_t} \frac{1}{\tilde{n}} \sum_{i=1}^{\tilde{n}} \rho_i \ell(W\phi_{t-1}(\mathbf{x}_{t,i}) + \eta_t Wg(\mathbf{x}_{t,i}), y_{t,i})\right]$$
$$+ \frac{4c\sigma_{\max}BLt^{1-s}}{1-s}\sqrt{\frac{\log 2/\delta}{\tilde{n}}}.$$

We now focus on bounding the Rademacher complexity term appearing above. To this end, we rely on the composition property of Rademacher complexity. Since $\ell$ is $L$-Lipscthiz in the first argument, applying Theorem I.3 we get

$$R(W_t, \phi_t) \leqslant R(W_{t-1}, \phi_t) + 2L\mathbb{E}\left[\sup_{W\in\mathcal{W}, g\in\mathcal{G}_t} \frac{1}{\tilde{n}} \sum_{i=1}^{\tilde{n}} \sum_{k=1}^{K} \rho_{ik} [W\phi_{t-1}(\mathbf{x}_{t,i}) + \eta_t Wg(\mathbf{x}_{t,i})]_k\right]$$
$$+ \frac{4c\sigma_{\max}BLt^{1-s}}{1-s}\sqrt{\frac{\log 2/\delta}{\tilde{n}}}$$
$$\leqslant R(W_{t-1}, \phi_t) + 2\eta_t LR(\mathcal{W}, \mathcal{G}_t) + 2L \underbrace{\mathbb{E}\left[\sup_{W\in\mathcal{W}} \frac{1}{\tilde{n}} \sum_{i=1}^{\tilde{n}} \sum_{k=1}^{K} \rho_{ik} [W\phi_{t-1}(\mathbf{x}_{t,i})]_k\right]}_{T_1}$$
$$+ \frac{4c\sigma_{\max}BLt^{1-s}}{1-s}\sqrt{\frac{\log 2/\delta}{\tilde{n}}}$$

$T_1$ can be bounded as follows. Let $\rho \in \mathbb{R}^{K\times\tilde{n}}$ be the matrix whose $(k,i)^{th}$ entry is given by $\rho_{ik}$ and $\phi_{t-1}(S_t) \in \mathbb{R}^{D\times\tilde{n}}$ be the matrix whose $(j,i)^{th}$ entry is given by $[\phi_{t-1}(\mathbf{x}_{t,i})]_j$. $T_1$ can be rewritten in terms of $\rho, \phi_{t-1}(S_t)$ as

$$T_1 = \mathbb{E}\left[\sup_{W\in\mathcal{W}} \frac{1}{\tilde{n}} \left\langle \rho\phi_{t-1}(S_t)^T, W\right\rangle_F\right]$$
$$\leqslant \left[\sup_{W\in\mathcal{W}} \|W\|_2\right] \mathbb{E}\left[\frac{1}{\tilde{n}} \|\rho\phi_{t-1}(S_t)^T\|_F\right]$$
$$\leqslant \sigma_{\max}\mathbb{E}\left[\frac{1}{\tilde{n}} \|\rho\phi_{t-1}(S_t)^T\|_F\right]$$
$$\leqslant \frac{\sigma_{\max}}{\tilde{n}}\sqrt{\mathbb{E}\left[\|\rho\phi_{t-1}(S_t)^T\|_F^2\right]}$$
$$= \frac{\sigma_{\max}}{\tilde{n}}\sqrt{K\mathbb{E}\left[\|\phi_{t-1}(S_t)\|_F^2\right]} \leqslant \sigma_{\max}\sqrt{\frac{K}{\tilde{n}}}\mathbb{E}\left[\sup_X \|\phi_{t-1}(X)\|_2\right]$$
$$\leqslant \frac{c\sigma_{\max}Bt^{1-s}}{1-s}\sqrt{\frac{K}{\tilde{n}}},$$

where the last inequality follows from our choice of step size $\eta_t$ and our assumption on the boundedness of the outputs of functions in $\mathcal{G}_t$. Substituting this upper bound on $T_1$ in the previous inequality gives us the required bound on $R(W_t, \phi_t)$. □

**Lemma E.2.** *Consider the setting of Theorem 4.1. Let $(W_t, \phi_t)$ be the $t^{th}$ iterate generated by Algorithm 1 with Algorithm 3 as update routine. Then for any $t$, the following holds with probability at least $1-2\delta$ over datasets of size $n$*

$$\langle g_t, -\nabla_\phi R(W_{t-1}, \phi_{t-1})\rangle_P \geqslant \beta B\|\nabla_\phi R(W_{t-1}, \phi_{t-1})\|_P - \epsilon_t - 2\sigma_{max}LR(\mathcal{G}_t) - 4\sigma_{max}BL\sqrt{\frac{\log 2/\delta}{\tilde{n}}}.$$

*Proof.* Let $\hat{P}_{\tilde{n},t}$ be the empirical distribution of dataset $S_t$. Since $\mathcal{G}_t$ satisfies the $(\beta, \epsilon_t)$-weak learning condition w.r.t dataset $S_t$, we have

$$\left\langle g_t, -\nabla_\phi \widehat{R}_{S_t}(W_{t-1}, \phi_{t-1}) \right\rangle_{P_{\tilde{n},t}^X} \geq \beta B \|\nabla_\phi \widehat{R}_{S_t}(W_{t-1}, \phi_{t-1})\|_{P_{\tilde{n},t}^X} - \epsilon_t.$$

Consider the following lower bound for $\langle g_t, -\nabla_\phi R(W_{t-1}, \phi_{t-1})\rangle_P$

$$\langle g_t, -\nabla_\phi R(W_{t-1}, \phi_{t-1})\rangle_P \geq \underbrace{\left\langle g_t, -\nabla_\phi \widehat{R}_{S_t}(W_{t-1}, \phi_{t-1}) \right\rangle_{P_{\tilde{n},t}^X}}_{T_1}$$

$$- \underbrace{\left| \left\langle g_t, -\nabla_\phi \widehat{R}_{S_t}(W_{t-1}, \phi_{t-1}) \right\rangle_{P_{\tilde{n},t}^X} - \langle g_t, -\nabla_\phi R(W_{t-1}, \phi_{t-1})\rangle_P \right|}_{T_2}$$

We now lower bound each of the terms appearing the RHS of the above inequality. Similar to the proof of Lemma E.1, throughout the proof we condition on the past datasets $S_1, \ldots S_{t-1}$ and show that the Lemma holds for any choice of $S_1, \ldots S_{t-1}$.

**Bounding $T_1$.** Using the weak learning condition, $T_1$ can be lower bounded as

$$T_1 \geq \beta B \|\nabla_\phi \widehat{R}_{S_t}(W_{t-1}, \phi_{t-1})\|_{P_{\tilde{n},t}^X} - \epsilon_t.$$

Using triangle inequality, this can be further lower bounded as

$$T_1 \geq \beta B \|\nabla_\phi R(W_{t-1}, \phi_{t-1})\|_P - \beta B \left| \|\nabla_\phi R(W_{t-1}, \phi_{t-1})\|_P - \|\nabla_\phi \widehat{R}_{S_t}(W_{t-1}, \phi_{t-1})\|_{P_{\tilde{n},t}^X} \right| - \epsilon_t.$$

We now bound the middle term in the RHS using standard concentration inequalities. Define random variable $Z$ as

$$Z = W_{t-1}^T \nabla \ell(W_{t-1}\phi_{t-1}(X), Y),$$

for $(X, Y) \sim P$ and define $\mathbf{z}_{t,i}$ as

$$\mathbf{z}_{t,i} = W_{t-1}^T \nabla \ell(W_{t-1}\phi_{t-1}(\mathbf{x}_{t,i}), y_{t,i}),$$

where $\nabla \ell(u, y)$ denotes the gradient of $\ell$ w.r.t its first argument. Then from the definition of functional gradients $\nabla_\phi \widehat{R}_{S_t}(W_{t-1}, \phi_{t-1}), \nabla_\phi R(W_{t-1}, \phi_{t-1})$, we have

$$\|\nabla_\phi \widehat{R}_{S_t}(W_{t-1}, \phi_{t-1})\|_{P_{\tilde{n},t}^X}^2 = \frac{1}{\tilde{n}} \sum_{i=1}^{\tilde{n}} \|\mathbf{z}_{t,i}\|^2, \quad \|\nabla_\phi R(W_{t-1}, \phi_{t-1})\|_P^2 = \mathbb{E}\left[\|Z\|^2\right].$$

Since $\ell$ is $L$-Lipschitz, it is easy to see that $\|Z\|$ is a bounded random variable and always lies in the interval $[0, \sigma_{\max} L]$. So using Chernoff bounds in Theorem I.1, we can show that the following holds with probability at least $1 - \delta$

$$\left| \sum_{i=1}^{\tilde{n}} \frac{1}{\tilde{n}} \|\mathbf{z}_{t,i}\|^2 - \mathbb{E}\left[\|Z\|^2\right] \right| \leq \sigma_{\max} L \sqrt{\frac{3\mathbb{E}\left[\|Z\|^2\right] \log 1/\delta}{\tilde{n}}}.$$

Now, consider the following

$$\left| \|\nabla_\phi R(W_{t-1}, \phi_{t-1})\|_P - \|\nabla_\phi \widehat{R}_{S_t}(W_{t-1}, \phi_{t-1})\|_{P_{\tilde{n},t}^X} \right| = \left| \sqrt{\sum_{i=1}^{\tilde{n}} \frac{1}{\tilde{n}} \|\mathbf{z}_{t,i}\|^2} - \sqrt{\mathbb{E}\left[\|Z\|^2\right]} \right|$$

$$\leq \frac{\left| \sum_{i=1}^{\tilde{n}} \frac{1}{\tilde{n}} \|\mathbf{z}_{t,i}\|^2 - \mathbb{E}\left[\|Z\|^2\right] \right|}{\sqrt{\mathbb{E}\left[\|Z\|^2\right]}},$$

where the last inequality follows from the fact that $|\sqrt{a} - \sqrt{b}| = \frac{|a-b|}{\sqrt{a}+\sqrt{b}} \leq \frac{|a-b|}{\sqrt{b}}$. This shows that, with probability at least $1 - \delta$, $T_1$ can be lower bounded as

$$T_1 \geq \beta B \|\nabla_\phi R(W_{t-1}, \phi_{t-1})\|_P - \beta \sigma_{\max} B L \sqrt{\frac{3 \log 1/\delta}{\tilde{n}}} - \epsilon_t. \tag{8}$$

**Bounding $T_2$.**  Using the definition of functional gradients, $T_2$ can be rewritten as follows

$$T_2 = \left| \mathbb{E}_X \left[ \langle g_t(X), \nabla_\phi R(W_{t-1}, \phi_{t-1})(X) \rangle \right] - \frac{1}{\tilde{n}} \sum_{i=1}^{\tilde{n}} \left\langle g_t(\mathbf{x}_{t,i}), \nabla_\phi \widehat{R}_{S_t}(W_{t-1}, \phi_{t-1})(\mathbf{x}_{t,i}) \right\rangle \right|$$

$$= \left| \mathbb{E}_{X,Y} \left[ \langle g_t(X), W_{t-1}^T \nabla \ell(W_{t-1}\phi_{t-1}(X), Y) \rangle \right] - \frac{1}{\tilde{n}} \sum_{i=1}^{\tilde{n}} \left\langle g_t(\mathbf{x}_{t,i}), W_{t-1}^T \nabla \ell(W_{t-1}\phi_{t-1}(\mathbf{x}_{t,i}), y_{t,i}) \right\rangle \right|$$

$$\leqslant \sup_{g \in \mathcal{G}_t} \left| \mathbb{E}_{X,Y} \left[ \langle g(X), W_{t-1}^T \nabla \ell(W_{t-1}\phi_{t-1}(X), Y) \rangle \right] - \frac{1}{\tilde{n}} \sum_{i=1}^{\tilde{n}} \left\langle g(\mathbf{x}_{t,i}), W_{t-1}^T \nabla \ell(W_{t-1}\phi_{t-1}(\mathbf{x}_{t,i}), y_{t,i}) \right\rangle \right|.$$

We now rely on uniform convergence bounds and bound the RHS in terms of the Rademacher complexity term $\mathcal{R}(\mathcal{G}_t)$. First note that the random variable $\langle g(X), W_{t-1}^T \nabla \ell(W_{t-1}\phi_{t-1}(X), Y) \rangle$ is bounded and lies in the interval $[-\sigma_{\max}BL, \sigma_{\max}BL]$. This follows from the Lipschitz property of the loss $\ell$ and the boundedness of the functions in $\mathcal{G}_t$. Using Theorem I.2, we get the following upper bound for $T_2$, which holds with probability at least $1 - \delta$

$$T_2 \leqslant 2\mathbb{E} \left[ \sup_{g \in \mathcal{G}_t} \frac{1}{\tilde{n}} \sum_{i=1}^{\tilde{n}} \rho_i \langle g(\mathbf{x}_{t,i}), W_{t-1}^T \nabla \ell(W_{t-1}\phi_{t-1}(\mathbf{x}_{t,i}), y_{t,i}) \rangle \right] + 2\sigma_{\max}BL\sqrt{\frac{\log 2/\delta}{\tilde{n}}}.$$

We now focus on bounding the Rademacher complexity term in the above inequality. Define function $h_i : \mathbb{R}^D \to \mathbb{R}$ as follows

$$h_i(\mathbf{u}) = \left\langle \mathbf{u}, W_{t-1}^T \nabla \ell(W_{t-1}\phi_{t-1}(\mathbf{x}_{t,i}), y_{t,i}) \right\rangle.$$

Note that, $h_i(\mathbf{u})$ is $\sigma_{\max}L$-Lipschitz in $\mathbf{u}$. The Rademacher complexity can be written in terms of $h_i$'s as follows

$$\mathbb{E} \left[ \sup_{g \in \mathcal{G}_t} \frac{1}{\tilde{n}} \sum_{i=1}^{\tilde{n}} \rho_i \langle g(\mathbf{x}_{t,i}), W_{t-1}^T \nabla \ell(W_{t-1}\phi_{t-1}(\mathbf{x}_{t,i}), y_{t,i}) \rangle \right] = \mathbb{E}_{S_t} \left[ \mathbb{E}_\rho \left[ \sup_{g \in \mathcal{G}_t} \frac{1}{\tilde{n}} \sum_{i=1}^{\tilde{n}} \rho_i h_i(g(\mathbf{x}_{t,i})) \middle| S_t \right] \right].$$

Using the composition property of Rademacher complexities stated in Theorem I.3, we get

$$\mathbb{E} \left[ \sup_{g \in \mathcal{G}_t} \frac{1}{\tilde{n}} \sum_{i=1}^{\tilde{n}} \rho_i \langle g(\mathbf{x}_{t,i}), W_{t-1}^T \nabla \ell(W_{t-1}\phi_{t-1}(\mathbf{x}_{t,i}), y_{t,i}) \rangle \right] \leqslant \sigma_{\max}L\mathbb{E}_{S_t} \left[ \mathbb{E}_\rho \left[ \sup_{g \in \mathcal{G}_t} \frac{1}{\tilde{n}} \sum_{i=1}^{\tilde{n}} \sum_{j=1}^{D} \rho_{ij}[g(\mathbf{x}_{t,i})]_j \middle| S_t \right] \right]$$

$$= \sigma_{\max}L\mathbb{E} \left[ \sup_{g \in \mathcal{G}_t} \frac{1}{\tilde{n}} \sum_{i=1}^{\tilde{n}} \sum_{j=1}^{D} \rho_{ij}[g(\mathbf{x}_{t,i})]_j \right]$$

$$= \sigma_{\max}L\mathcal{R}(\mathcal{G}_t).$$

So we have the following bound for $T_2$ which holds with probability at least $1 - \delta$

$$T_2 \leqslant 2\sigma_{\max}L\mathcal{R}(\mathcal{G}_t) + 2\sigma_{\max}BL\sqrt{\frac{\log 2/\delta}{\tilde{n}}}. \tag{9}$$

Combining Equations (8), (9) gives us the required bound.  □

## E.2   Main Argument

Our analysis of inexact gradient descent uses similar arguments as in Temlyakov [34]. Let $\phi_t = \phi_{t-1} + \eta_t g_t$ be the $t^{th}$ iterate generated by the algorithm. We first derive an upper bound for the reduction in population risk in the $t^{th}$ iteration of the algorithm. From Lemma E.1 we know that with probability at least $1 - \delta/3T$

$$R(W_t, \phi_t) \leqslant R(W_{t-1}, \phi_t) + C_1(t), \tag{10}$$

where $C_1(t) = 2\eta_t L\mathcal{R}(\mathcal{W}, \mathcal{G}_t) + \frac{4c\sigma_{\max}BLt^{1-s}}{1-s}\left(\sqrt{\frac{\log 6T/\delta}{\tilde{n}}} + \sqrt{\frac{K}{\tilde{n}}}\right)$. Since $\ell$ is $M$ smooth, the following holds for any two vectors $\mathbf{u}, \mathbf{v} \in \mathbb{R}^K$ and $y \in \mathcal{Y}$

$$\ell(\mathbf{u} + \mathbf{v}, y) \leqslant \ell(\mathbf{u}, y) + \langle \mathbf{v}, \nabla \ell(\mathbf{u}, y) \rangle + \frac{M\|\mathbf{v}\|_2^2}{2}.$$

Using this smoothness property, $R(W_{t-1}, \phi_t) = \mathbb{E}\left[\ell(W_{t-1}\phi_{t-1}(\mathbf{x}) + \eta_t W_{t-1}g_t, y)\right]$ can be upper bounded as

$$R(W_{t-1}, \phi_t) \leqslant R(W_{t-1}, \phi_{t-1}) + \eta_t \langle g_t, \nabla_\phi R(W_{t-1}, \phi_{t-1})\rangle_P + \frac{\eta_t^2 M \sigma_{\max}^2 \|g_t\|_P^2}{2}. \tag{11}$$

Combining Equations (10), (11), we get the following bound on $R(W_t, \phi_t)$ which holds with probability at least $1 - \delta/3T$

$$R(W_t, \phi_t) \leqslant R(W_{t-1}, \phi_{t-1}) + \eta_t \langle g_t, \nabla_\phi R(W_{t-1}, \phi_{t-1})\rangle_P + \frac{\eta_t^2 M \sigma_{\max}^2 B^2}{2} + C_1(t).$$

Next, from Lemma E.2 we know that the $g_t$ chosen by the algorithm satisfies the following with probability at least $1 - 2\delta/3T$

$$\langle g_t, -\nabla_\phi R(W_{t-1}, \phi_{t-1})\rangle_P \geqslant \beta B \|\nabla_\phi R(W_{t-1}, \phi_{t-1})\|_P - \epsilon_t - C_2(t),$$

where $C_2(t) = 2\sigma_{\max} L \mathcal{R}(\mathcal{G}_t) + 4\sigma_{\max} BL\sqrt{\frac{\log 6T/\delta}{\tilde{n}}}$. Substituting this in the previous equation, we get the following bound on $R(W_t, \phi_t)$ which holds with probability at least $1 - \delta/T$

$$R(W_t, \phi_t) \leqslant R(W_{t-1}, \phi_{t-1}) - \eta_t \beta B \|\nabla_\phi R(W_{t-1}, \phi_{t-1})\|_P + \frac{c^2 M B^2 \sigma_{\max}^2}{2} t^{-2s} \tag{12}$$

$$+ \eta_t \epsilon_t + C_1(t) + \eta_t C_2(t). \tag{13}$$

Let $r_t = R(W_t, \phi_t) - R(W^*, \phi^*) - \sum_{i=1}^{t}(\eta_i \epsilon_i + C_1(t) + \eta_t C_2(t))$. Then the above equation implies the following recurrence on $r_t$

$$r_t \leqslant r_{t-1} + \frac{c^2 M B^2 \sigma_{\max}^2}{2} t^{-2s}. \tag{14}$$

We now try to tighten this recurrence. Let $W_{t-1}^\dagger$ be the pseudoinverse of $W_{t-1}$. From the convexity of $\ell$ we have

$$R(W_{t-1}, \phi_{t-1}) - R(W^*, \phi^*) \overset{(a)}{=} R(W_{t-1}, \phi_{t-1}) - R(W_{t-1}, W_{t-1}^\dagger W^* \phi^*)$$

$$\overset{(b)}{\leqslant} -\left\langle W_{t-1}^\dagger W^* \phi^* - \phi_{t-1}, \nabla_\phi R(W_{t-1}, \phi_{t-1})\right\rangle_P$$

$$\leqslant \|\nabla_\phi R(W_{t-1}, \phi_{t-1})\|_P \left(\sigma_{\min}^{-1} \|W^*\|_2 \|\phi^*\|_P + \|\phi_{t-1}\|_P\right),$$

where $(a)$ follows from the definition of psuedoinverse and $(b)$ follows from the convexity of $\ell$. Letting $A_t = \sum_{i=1}^{t} \eta_i B$, we can lower bound $\|\nabla_\phi R(W_{t-1}, \phi_{t-1})\|$ as

$$\|\nabla_\phi R(W_{t-1}, \phi_{t-1})\| \geqslant \frac{R(W_{t-1}, \phi_{t-1}) - R(W^*, \phi^*)}{\sigma_{\min}^{-1} \|W^*\|_2 \|\phi^*\|_P + A_{t-1}}.$$

Substituting this in Equation (12), we get

$$R(W_t, \phi_t) \leqslant R(W_{t-1}, \phi_{t-1}) - \eta_t \beta B \left(\frac{R(W_{t-1}, \phi_{t-1}) - R(W^*, \phi^*)}{\sigma_{\min}^{-1} \|W^*\|_2 \|\phi^*\|_P + A_{t-1}}\right) + \frac{c^2 M B^2 \sigma_{\max}^2}{2} t^{-2s}$$

$$+ \eta_t \epsilon_t + C_1(t) + \eta_t C_2(t).$$

Rewriting the above equation in terms of $r_t$, we get

$$
\begin{aligned}
r_t &\leqslant r_{t-1} - \eta_t \beta B \left(\frac{r_{t-1}}{\sigma_{\min}^{-1}\|W^*\|_2\|\phi^*\|_P + A_{t-1}}\right) + \frac{c^2 M B^2 \sigma_{\max}^2}{2} t^{-2s} \\
&= \left(1 - \frac{\eta_t \beta B}{\sigma_{\min}^{-1}\|W^*\|_2\|\phi^*\|_P + A_{t-1}}\right) r_{t-1} + \frac{c^2 M B^2 \sigma_{\max}^2}{2} t^{-2s}.
\end{aligned}
\tag{15}
$$

In the rest of the proof, we try to solve the above recurrence relation on $r_t$ to obtain the required excess risk bound. First note that there exists $t_0$ such that for all $t \geqslant t_0$ [2]

$$\frac{\eta_t \beta B}{\sigma_{\min}^{-1} \|W^*\|_2 \|\phi^*\|_P + A_{t-1}} \geqslant \frac{\alpha + 3\beta(1-s)}{4t}. \tag{16}$$

This follows from the observation that $\eta_t = ct^{-s}$ and $A_{t-1} \leqslant \frac{cBt^{1-s}}{1-s}$. We now make use of Theorem I.5 for solving the recurrence in Equation (15). We first show that $r_t$ satisfies the conditions for Theorem I.5 with $a = \alpha, b = (\alpha + \beta(1-s))/2$ and $D = t_0$ and for some $A$ which we specify later. From Equation (14) we have

$$r_{t+1} \leqslant r_t + \frac{c^2 MB^2 \sigma_{\max}^2}{2} t^{-2s} \leqslant r_t + A(t-1)^{-\alpha},$$

where the last inequality holds for any $A \geqslant \frac{c^2 MB^2 \sigma_{\max}^2}{2}$ and for our choice of $\alpha, s$ specified in the theorem statement. This shows that the first condition of Theorem I.5 is satisfied by $r_t$. Next, suppose $r_t \geqslant At^{-\alpha}$, for some $t \geqslant t_0$. Then using Equations (15) and (16), $r_{t+1}$ can be bounded as follows

$$r_{t+1} \leqslant \left(1 - \frac{\alpha + 3\beta(1-s)}{4t}\right) r_t + \frac{c^2 MB^2 \sigma_{\max}^2}{2} t^{-2s}$$

$$= \left(1 - \frac{\alpha + \beta(1-s)}{2t}\right) r_t \underbrace{- \left(\frac{\beta(1-s) - \alpha}{4t}\right) r_t + \frac{c^2 MB^2 \sigma_{\max}^2}{2} t^{-2s}}_{T_1}.$$

Following our choices for $\alpha, s$ and using the fact that $r_t \geqslant At^{-\alpha}$, it is easy to verify that $T_1 \leqslant 0$ for sufficiently large $A$. This shows that for appropriately chosen $A$, we have

$$r_{t+1} \leqslant \left(1 - \frac{\alpha + \beta(1-s)}{2t}\right) r_t.$$

Since the conditions for Theorem I.5 are satisfied, using it to solve our recurrence gives us the following bound on $r_t$ which holds with probability at least $1 - \delta$

$$r_T \leqslant O\left(\frac{1}{T^\alpha}\right).$$

This finishes the proof of the Theorem.

## F   Proof of Corollary 4.1

A simple intuition for why the exact greedy approach satisfies similar risk bounds as gradient greedy approach is that in exact greedy approach one solves the greedy step in Equation (2) exactly. Whereas, in gradient greedy approach, the greedy step is only solved approximately and so one would expect the objective value of exact greedy approach to be smaller than gradient greedy approach. We formalize this intuition in the proof. Let $(W_t, \phi_t)$, where $\phi_t = \phi_{t-1} + \eta_t g_t$, be the $t^{th}$ iterate generated by the exact greedy algorithm. And let $(\tilde{W}, \tilde{\phi}_t)$, where $\tilde{\phi}_t = \phi_{t-1} + \eta_t \tilde{g}_t$, be the iterate obtained by running gradient greedy update in the $t^{th}$ iteration of the algorithm. We now bound $R(W_t, \phi_t)$ in terms of $R(\tilde{W}, \tilde{\phi}_t)$

$$R(W_t, \phi_t) \leqslant \widehat{R}_{S_t}(W_t, \phi_t) + \sup_{W \in \mathcal{W}, g \in \mathcal{G}_t} |R(W, \phi_{t-1} + \eta_t g) - \widehat{R}_{S_t}(W, \phi_{t-1} + \eta_t g)|$$

$$\overset{(a)}{\leqslant} \widehat{R}_{S_t}(\tilde{W}_t, \tilde{\phi}_t) + \sup_{W \in \mathcal{W}, g \in \mathcal{G}_t} |R(W, \phi_{t-1} + \eta_t g) - \widehat{R}_{S_t}(W, \phi_{t-1} + \eta_t g)|$$

$$\leqslant R((\tilde{W}_t, \tilde{\phi}_t)) + 2 \sup_{W \in \mathcal{W}, g \in \mathcal{G}_t} |R(W, \phi_{t-1} + \eta_t g) - \widehat{R}_{S_t}(W, \phi_{t-1} + \eta_t g)|,$$

where $(a)$ follows from the definition of $W_t, \phi_t$ which are obtained by minimizing Equation (2). Note that the supremum in the RHS above can be bounded using Lemma E.1.

From the proof of Theorem 4.1, we know that $R((\tilde{W}_t, \tilde{\phi}_t))$ can be upper bounded in terms of $R((W_{t-1}, \phi_{t-1}))$. To be precise, from Equation (12) in the proof of Theorem 4.1, we know that with probability at least $1 - 2\delta/T$

$$R(\tilde{W}_t, \tilde{\phi}_t) \leqslant R(W_{t-1}, \phi_{t-1}) - \eta_t \beta B \|\nabla_\phi R(W_{t-1}, \phi_{t-1})\|_P + \frac{c^2 MB^2 \sigma_{\max}^2}{2} t^{-2s}$$

$$+ \eta_t \epsilon_t + C_1(t) + \eta_t C_2(t).$$

This shows that

$$R(W_t, \phi_t) \leqslant R(W_{t-1}, \phi_{t-1}) - \eta_t \beta B \|\nabla_\phi R(W_{t-1}, \phi_{t-1})\|_P + \frac{c^2 M B^2 \sigma_{\max}^2}{2} t^{-2s}$$
$$+ \eta_t \epsilon_t + C_1(t) + \eta_t C_2(t).$$

Using the exact same techniques as in the proof of Theorem 4.1, we get the required risk bound on $R(W_T, \phi_T)$.

# G    Proof of Corollary 4.2

The major part of the proof involves bounding the Rademacher complexity terms appearing in the risk bound of Theorem 4.1. We first bound $\mathcal{R}(\mathcal{G})$.

$$\mathcal{R}(\mathcal{G}) = \mathbb{E}\left[\sup_{g \in \mathcal{G}} \frac{1}{\tilde{n}} \sum_{i=1}^{\tilde{n}} \sum_{j=1}^{D} \rho_{ij}[g(\mathbf{x}_{t,i})]_j\right]$$

$$= \mathbb{E}\left[\sup_{C:\max_j \|C_{j,*}\|_1 \leqslant \Lambda} \frac{1}{\tilde{n}} \sum_{i=1}^{\tilde{n}} \sum_{j=1}^{D} \rho_{ij}\sigma(\langle C_{j,*}, \mathbf{x}_{t,i}\rangle)\right]$$

$$\leqslant \sum_{j=1}^{D} \mathbb{E}\left[\sup_{\|C_{j,*}\|_1 \leqslant \Lambda} \frac{1}{\tilde{n}} \sum_{i=1}^{\tilde{n}} \rho_{ij}\sigma(\langle C_{j,*}, \mathbf{x}_{t,i}\rangle)\right]$$

$$\overset{(a)}{\leqslant} \sum_{j=1}^{D} \mathbb{E}\left[\sup_{\|C_{j,*}\|_1 \leqslant \Lambda} \frac{1}{\tilde{n}} \sum_{i=1}^{\tilde{n}} \rho_{ij} \langle C_{j,*}, \mathbf{x}_{t,i}\rangle\right]$$

$$\leqslant \sum_{j=1}^{D} \Lambda \mathbb{E}\left[\frac{1}{\tilde{n}} \left\|\sum_{i=1}^{\tilde{n}} \rho_{ij}\mathbf{x}_{t,i}\right\|_\infty\right]$$

$$= \frac{D\Lambda}{\tilde{n}} \mathbb{E}\left[\left\|\sum_{i=1}^{\tilde{n}} \rho_{i1}\mathbf{x}_{t,i}\right\|_\infty\right]$$

$$\overset{(b)}{\leqslant} 2D\Lambda\sqrt{\frac{\log d}{\tilde{n}}},$$

where $(a)$ follows from the Lipschitzness of sigmoid activation function and composition property of Rademacher complexities(see Theorem I.3) and $(b)$ follows from the following well known property of sub-Gaussian random variables. Let $Z_1, \ldots Z_n$ be $n$ random variables, not necessarily independent. Moreover, lets suppose each $Z_i$ is sub-Gaussian with parameter $\sigma$. Then $\mathbb{E}\left[\max_i Z_i\right] \leqslant \sqrt{2\sigma^2 \log n}$. Since $\mathcal{X} \subseteq [0,1]^d$, it is easy to see that conditioned on data $S_t$, each co-ordinate of $\sum_{i=1}^{\tilde{n}} \rho_{i1}\mathbf{x}_{t,i}$ is a sub-Gaussian random variable with parameter $\sqrt{\tilde{n}}$. So using the above stated property of sub-Gaussian random variables, we get

$$\mathbb{E}_\rho\left[\left\|\sum_{i=1}^{\tilde{n}} \rho_{i1}\mathbf{x}_{t,i}\right\|_\infty\right] = \mathbb{E}_\rho\left[\max_{j \in [d]} \max\left\{\sum_{i=1}^{\tilde{n}} \rho_{i1}[\mathbf{x}_{t,i}]_j, -\sum_{i=1}^{\tilde{n}} \rho_{i1}[\mathbf{x}_{t,i}]_j\right\}\right]$$
$$\leqslant \sqrt{2\tilde{n}\log 2d}.$$

Next, we bound $\mathcal{R}(\mathcal{W}, \mathcal{G})$

$$
\begin{aligned}
\mathcal{R}(\mathcal{W}, \mathcal{G}) &= \mathbb{E}\left[\sup_{\substack{W \in \mathcal{W}, \\ g \in \mathcal{G}}} \frac{1}{\tilde{n}} \sum_{i=1}^{\tilde{n}} \sum_{k=1}^{K} \rho_{ik}[W g(\mathbf{x}_{t,i})]_k\right] \\
&\leqslant \sum_{k=1}^{K} \mathbb{E}\left[\sup_{\substack{W \in \mathcal{W}, \\ g \in \mathcal{G}}} \frac{1}{\tilde{n}} \sum_{i=1}^{\tilde{n}} \rho_{ik}\langle W_{k,*}, g(\mathbf{x}_{t,i})\rangle\right] \\
&\overset{(a)}{\leqslant} 2\sigma_{\max} K \sum_{j=1}^{D} \mathbb{E}\left[\sup_{\|C_{j,*}\|_1 \leqslant \Lambda} \frac{1}{\tilde{n}} \sum_{i=1}^{\tilde{n}} \rho_i \langle C_{j,*}, \mathbf{x}_{t,i}\rangle\right] + O\left(\frac{\sigma_{\max} K \sqrt{D} \log D}{\sqrt{\tilde{n}}}\right) \\
&\overset{(b)}{\leqslant} 4\sigma_{\max} K D \Lambda \sqrt{\frac{\log d}{\tilde{n}}} + O\left(\frac{\sigma_{\max} K \sqrt{D} \log D}{\sqrt{\tilde{n}}}\right) \\
&\leqslant O\left(\frac{\sigma_{\max} K D \Lambda \log(dD)}{\sqrt{\tilde{n}}}\right).
\end{aligned}
$$

where $(a)$ follows from the property of Rademacher complexity stated in Theorem I.4 and $(b)$ uses the arguments used to bound $\mathcal{R}(\mathcal{G})$ above. Substituting the above bounds for $\mathcal{R}(\mathcal{G})$ and $\mathcal{R}(\mathcal{W}, \mathcal{G})$ in Theorem 4.1 and using the fact that $\sup_X \|g(X)\|_2 \leqslant \sqrt{D}$, for all $g \in \mathcal{G}$, we get the required risk bound.

## H  Proof of Corollary 4.3

Similar to the proof of Corollary 4.2, we focus on bounding the Radmacher complexity terms $\mathcal{R}(\mathcal{G}_t)$ and $\mathcal{R}(\mathcal{W}, \mathcal{G}_t)$. To bound $\mathcal{R}(\mathcal{G}_t)$, we use the same argument we used to bound $\mathcal{R}(\mathcal{G})$ in Corollary 4.2.

$$
\begin{aligned}
\mathcal{R}(\mathcal{G}_t) &= \mathbb{E}\left[\sup_{g \in \mathcal{G}_t} \frac{1}{\tilde{n}} \sum_{i=1}^{\tilde{n}} \sum_{j=1}^{D} \rho_{ij}[g(\mathbf{x}_{t,i})]_j\right] \\
&= \mathbb{E}\left[\sup_{C:\max_j \|C_{j,*}\|_1 \leqslant \Lambda} \frac{1}{\tilde{n}} \sum_{i=1}^{\tilde{n}} \sum_{j=1}^{D} \rho_{ij}\sigma(\langle C_{j,*}, \phi_{t-1}(\mathbf{x}_{t,i})\rangle)\right] \\
&\leqslant \sum_{j=1}^{D} \mathbb{E}\left[\sup_{\|C_{j,*}\|_1 \leqslant \Lambda} \frac{1}{\tilde{n}} \sum_{i=1}^{\tilde{n}} \rho_{ij}\sigma(\langle C_{j,*}, \phi_{t-1}(\mathbf{x}_{t,i})\rangle)\right] \\
&\leqslant \sum_{j=1}^{D} \mathbb{E}\left[\sup_{\|C_{j,*}\|_1 \leqslant \Lambda} \frac{1}{\tilde{n}} \sum_{i=1}^{\tilde{n}} \rho_{ij}\langle C_{j,*}, \phi_{t-1}(\mathbf{x}_{t,i})\rangle\right] \\
&\leqslant \sum_{j=1}^{D} \Lambda \mathbb{E}\left[\frac{1}{\tilde{n}}\left\|\sum_{i=1}^{\tilde{n}} \rho_{ij}\phi_{t-1}(\mathbf{x}_{t,i})\right\|_\infty\right] \\
&= \frac{D\Lambda}{\tilde{n}} \mathbb{E}\left[\left\|\sum_{i=1}^{\tilde{n}} \rho_i \phi_{t-1}(\mathbf{x}_{t,i})\right\|_\infty\right] \\
&\overset{(a)}{\leqslant} \frac{2cD\Lambda t^{1-s}}{1-s}\sqrt{\frac{\log d}{\tilde{n}}},
\end{aligned}
$$

where $(a)$ uses similar arguments as in the proof of Corollary 4.2 and relies on the fact that $\|\phi_{t-1}(\mathbf{x})\|_\infty \leqslant \sum_{i=1}^{t-1} \eta_i \leqslant \frac{ct^{1-s}}{1-s}$. Next, we bound $\mathcal{R}(\mathcal{W}, \mathcal{G}_t)$

$$
\begin{aligned}
\mathcal{R}(\mathcal{W}, \mathcal{G}_t) &= \mathbb{E}\left[\sup_{\substack{W \in \mathcal{W}, \\ g \in \mathcal{G}_t}} \frac{1}{\tilde{n}} \sum_{i=1}^{\tilde{n}} \sum_{k=1}^{K} \rho_{ik}[Wg(\mathbf{x}_{t,i})]_k\right] \\
&\leqslant \sum_{k=1}^{K} \mathbb{E}\left[\sup_{\substack{W \in \mathcal{W}, \\ g \in \mathcal{G}_t}} \frac{1}{\tilde{n}} \sum_{i=1}^{\tilde{n}} \rho_{ik} \langle W_{k,*}, g(\mathbf{x}_{t,i})\rangle\right] \\
&\overset{(a)}{\leqslant} 2\sigma_{\max} K \sum_{j=1}^{D} \mathbb{E}\left[\sup_{\|C_{j,*}\|_1 \leqslant \Lambda} \frac{1}{\tilde{n}} \sum_{i=1}^{\tilde{n}} \rho_i \langle C_{j,*}, \phi_{t-1}(\mathbf{x}_{t,i})\rangle\right] + O\left(\frac{\sigma_{\max} K \sqrt{D} \log D}{\sqrt{\tilde{n}}}\right) \\
&\overset{(b)}{\leqslant} \frac{4\sigma_{\max} cDK\Lambda t^{1-s}}{1-s} \sqrt{\frac{\log d}{\tilde{n}}} + O\left(\frac{\sigma_{\max} K \sqrt{D} \log D}{\sqrt{\tilde{n}}}\right) \\
&\leqslant O\left(t^{1-s} \frac{\sigma_{\max} K D\Lambda \log dD}{\sqrt{\tilde{n}}}\right),
\end{aligned}
$$

where $(a)$ follows from the property of Rademacher complexity stated in Theorem I.4 and $(b)$ relies on arguments used to bound $\mathcal{R}(\mathcal{G}_t)$. Substituting the above bounds for $\mathcal{R}(\mathcal{G}_t)$ and $\mathcal{R}(\mathcal{W}, \mathcal{G}_t)$ in Theorem 4.1, we get the required risk bound.

# I  Some Useful Results

**Theorem I.1** (Chernoff Bounds). *Let $X = \sum_{i=1}^{n} X_i$, where $X_i$'s are independently distributed in $[0,1]$. Then, for $\epsilon \in (0,1)$*

$$
\mathbb{P}\left(X > (1+\epsilon)\mathbb{E}\left[X\right]\right) \leqslant \exp\left(-\frac{\epsilon^2}{3}\mathbb{E}\left[X\right]\right), \quad \mathbb{P}\left(X < (1-\epsilon)\mathbb{E}\left[X\right]\right) \leqslant \exp\left(-\frac{\epsilon^2}{2}\mathbb{E}\left[X\right]\right).
$$

**Theorem I.2** (Bartlett and Mendelson [4]). *Let $\mathcal{F}$ be a class of functions mapping $\mathcal{X}$ to $[a,b]$ and let $\{X_i\}_{i=1}^{n}$ be independently selected according to the probability measure $P$. Then for any integer $n$ and any $0 < \delta < 1$, with probability at least $1 - \delta$ over samples of length $n$, every $f$ in $\mathcal{F}$ satisfies*

$$
\left|\frac{1}{n} \sum_{i=1}^{n} f(X_i) - \mathbb{E}\left[f(X)\right]\right| \leqslant 2\mathcal{R}(\mathcal{F}) + (b-a)\sqrt{\frac{\log 2/\delta}{n}},
$$

*where $\mathcal{R}(\mathcal{F})$ is the Rademacher complexity of $\mathcal{F}$ which is defined as*

$$
\mathcal{R}(\mathcal{F}) = \mathbb{E}\left[\sup_{f \in \mathcal{F}} \frac{1}{n} \sum_{i=1}^{n} \rho_i f(X_i)\right],
$$

*where the expectation is taken w.r.t the Rademacher random variables $\rho$'s and data $\{X_i\}_{i=1}^{n}$.*

We next present an important result on the composition property of Rademacher complexities.

**Theorem I.3** (Maurer [27]). *Let $\mathcal{F}$ be a class of functions mapping $\mathcal{X}$ to $\mathbb{R}^d$ and let $\{h_i\}_{i=1}^{n}$ be $L$-Lipschitz functions from $\mathbb{R}^d$ to $\mathbb{R}$. Then*

$$
\mathbb{E}_\rho\left[\sup_{f \in \mathcal{F}} \frac{1}{n} \sum_{i=1}^{n} \rho_i h_i(f(X_i))\right] \leqslant L\mathbb{E}_\rho\left[\sup_{f \in \mathcal{F}} \frac{1}{n} \sum_{i=1}^{n} \sum_{j=1}^{d} \rho_{ij}[f(X_i)]_j\right].
$$

**Theorem I.4** (Proposition A.12 of Allen-Zhu et al. [1]). *Let $u : \mathbb{R} \to \mathbb{R}$ be a fixed 1-Lipschitz function. Given $\mathcal{F}_1 \dots \mathcal{F}_m$ classes of functions $\mathcal{X} \to \mathbb{R}$ and suppose for each $j \in [m]$ there exists a function $f_j^{(0)} \in \mathcal{F}_j$ satisfying $\sup_{\mathbf{x} \in \mathcal{X}} |u(f_j^{(0)}(\mathbf{x}))| \leqslant A$, then*

$$
\mathcal{F}' = \left\{\mathbf{x} \to \sum_{j=1}^{m} v_j u(f_j(\mathbf{x})) \Big| f_j \in \mathcal{F}_j \wedge \|v\|_1 \leqslant B \wedge \|v\|_\infty \leqslant D\right\}
$$

*satisfies*

$$\mathbb{E}\left[\sup_{f\in\mathcal{F}'}\sum_{i=1}^{n}\frac{1}{n}\rho_i\sum_{j=1}^{m}v_j u(f_j(\mathbf{x}_i))\right] \leqslant 2D\sum_{j=1}^{m}\mathbb{E}\left[\sup_{f\in\mathcal{F}_j}\sum_{i=1}^{n}\frac{1}{n}\rho_i f(\mathbf{x}_i)\right] + O\left(\frac{AB\log m}{\sqrt{n}}\right).$$

**Theorem I.5** (Temlyakov [34]). *Let four positive numbers $a < b \leqslant 1$, $A, D \in \mathbb{N}$ be given and let a sequence $\{r_t\}_{t=1}^{\infty}$ have the following properties: $r_1 \leqslant A$ and for any $t \geqslant 2$*

$$r_t \leqslant r_{t-1} + A(t-1)^{-a}.$$

*Moreover, suppose the sequence is such that if $r_t \geqslant At^{-a}$ for some $t \geqslant D$, then $r_{t+1} \leqslant r_t(1 - b/t)$. Then there exists a constant $C$ such that for all $t \in \mathbb{N}$ we have*

$$r_t \leqslant Ct^{-a}.$$

# J   Experiments

This section provides experimental details, including the datasets, hyperparameter settings, and additional experimental evidence not presented in the main paper.

We first note that in our experiments for DenseCompBoost, we use a slight variant of $\mathcal{G}_t$ defined in Equation (3)

$$\mathcal{G}_t = \left\{ h + g \circ \left(\sum_{i=0}^{t-1}\alpha_i\phi_i\right), \text{ for } h \in \mathcal{H}, g \in \mathcal{G}, \alpha_i \in \mathbb{R} \right\},$$

where $\mathcal{H}, \mathcal{G}$ are weak feature transformer classes. We use this variant because the dimensions of the input feature space and the representation space need not be the same, and as a consequence $\mathcal{G}_t$ in Equation (3) can not always be used. Similar to StdCompBoost, we consider two choices for $\mathcal{H}, \mathcal{G}$: one based on fully connected blocks and the other based on convolution blocks.

## J.1   Drawbacks of Layer-by-Layer fitting

In this section, we provide empirical evidence highlighting drawbacks of layer-by-layer fitting and how our proposed techniques address these drawbacks. Similar to Section 5, we use StdCompBoost to denote standard layer-by-layer fitting.

**DenseCompBoost can recover from mistakes.**   We mentioned earlier that compared to StdComp-Boost, one advantage of DenseCompBoost is that the dense connections allow it to more easily recover from mistakes made in earlier layers. We now provide empirical evidence to support this claim. We introduce mistakes in the weights of the first layers learned using StdCompBoost and DenseCompBoost. To be precise, we fix the weights of the first layer of both StdCompBoost and DenseCompBoost to (a) the same random matrix, (b) an all-0 matrix, and then continue the training of the later layers. Table 3 shows the results: while StdCompBoost suffers a significant performance drop (from $82.49\%$ when every layer is greedily trained, to $72.99\%$ with a random first layer), the performance of dense greedy is barely affected (from $95.70\%$ when every layer is trained, to $95.0\%$ with a random first layer). Similar trend occurs when setting the first layer to 0: dense greedy still achieves a $93.69\%$ test accuracy, while standard greedy would fail to train at all since any signal in the data has been cut off.

|  |  | layer 1 | layer 2 | layer 3 | layer 4 | layer 5 |
|---|---|---|---|---|---|---|
| StdCompBoost | Random | 49.71 | 50.25 | 52.51 | 69.70 | 72.99 |
| DenseCompBoost | Random | 49.71 | 50.86 | 70.07 | 92.31 | 95.00 |
|  | Zero | 50.06 | 61.76 | 89.19 | 93.17 | 93.69 |

Table 3: Test accuracy at each layer, with the first layer being set to a random value or the all-0 matrix. Compared to the performance without corrupted first layer, StdCompBoost suffers a performance drop, while DenseCompBoost is almost unaffected, demonstrating its ability to recover from mistakes made in early layers.

**Narrow-to-Wide architecture of CmplxCompBoost.** Note that in CmplxCompBoost, we increase the widths of layers over iterations. We now justify this choice of architecture. There are two possible ways to vary the complexity of the $\tilde{\mathcal{G}}_t$, increasing or decreasing. We tested both approaches on one tabular dataset CovType, and one image dataset SVHN. On CovType, we started with a layer width of 4096, then increase or decrease the width of subsequent layers by 512 at each layer. On SVHN, the starting layer width is 128, followed by 4 additional layers, each increasing or decreasing the width by 16. As can be seen in table 4, increasing complexity gives slightly better results for both the datasets, therefore we choose to increase the width for CmplxCompBoost in all other experiments.

|  | Decreasing width | Increasing width |
|---|---|---|
| CovType | $95.58 \pm 0.04$ | $\mathbf{95.64} \pm 0.16$ |
| SVHN | $88.30 \pm 0.28$ | $\mathbf{89.05} \pm 0.01$ |

Table 4: Test accuracy using CmplxCompBoost with decreasing or increasing layer widths.

## J.2 Datasets and Hyperparameters

In this section, we present the details of datasets used in our experiments and describe our process for hyperparameter selection.

**Simulated Datasets.** We generated 3 synthetic binary classification datasets in $\mathbb{R}^{32}$. Simulation 1 is a concentric ellipsoids dataset, where a point $\mathbf{x}$ is classified based on $\mathbf{x}^T A \mathbf{x}$, for some randomly generated positive semi-definite matrix $A$. Simulations 2 and 3 are datasets whose classification boundaries are polynomials of degrees 8 and 9 respectively. For each of these datasets, we generated $10^6$ samples for training and testing.

*Hyper-parameters.* We used hold-out set validation to pick the best hyper-parameters for all the methods. We used $20\%$ of the training data as validation data and picked the best parameters using grid search, based on validation accuracy. After picking the best parameters, we train on the entire training data and report performance on the test data. For all the greedy techniques based on neural networks, we used fully connected blocks and tuned the following parameters: weight decay, width of weak feature transformers, number of iterations $T$. For CmplxCompBoost, we set $\Delta = D_0/5$. For end-to-end training, we tuned weight decay, width of layers, depth. We used SGD for optimization of all these techniques. The number of epochs and step size schedule of SGD are chosen to ensure convergence. For XGBoost, we tuned the number of trees, depth of each tree, learning rate.

**Benchmark Datasets.** We consider the following image datasets: CIFAR10, MNIST, FashionM-NIST [35], MNIST-rot-back-image [24], convex [35], SVHN [28], and the following tabular datasets from UCI repository [7]: letter recognition [17], forest cover type (covtype), connect4. The convex dataset involves classifying shapes in images as either convex or non-convex. MNIST-rot-back-image is generated from MNIST by rotating the images and adding random images in the background.

*Hyper-parameters.* For covtype dataset, which doesn't come with a test set, we randomly sample $20\%$ of the original data and use it as the test set. We use a similar hyper-parameter selection technique as above and tune the same set of hyper-parameters as described above. We use convolution blocks for CIFAR10, SVHN, FashionMNIST, convex, MNIST-rot-back-image and fully connected blocks for the rest. We limit the width of fully connected blocks to 4096, and the number of output channels in convolution blocks to 128 while tuning the hyper-parameters for the composition boosting techniques and end-to-end training. For AdaBoost and additive representation boosting, we set these limits to 16000 and 350 respectively. For CmplxCompBoost with convolution blocks, we set $\Delta = D_0/8$. We *do not* use data augmentation in our experiments.

## J.3 Further Experimental Details

Tables 5, 6 list the statistics of datasets used in our experiments. We now list the hyper-parameters tuned for each dataset and learning algorithm. Table 7 presents the list of hyper-parameters tuned for XGBoost. All the other techniques we use in our experiments rely on neural networks. We use SGD with momentum to learn these models. In all our experiments, we set the initial learning rate of SGD to 0.01, momentum to 0.9, batch size to 64 and tune the following weight decay values: $\{0.0001, 0.0005, 0.001, 0.005, 0.01\}$. The number of epochs we used for SGD varied with the dataset and is chosen to be large enough to ensure convergence. Over the course of the SGD optimization,

we reduce the learning rate by a factor of $0.5$, if the training loss doesn't decrease for certain number of SGD iterations (we rely on scheduler-tolerance option in PyTorch to implement this). We run all the greedy techniques (*AdaBoost, additive feature boosting, StdCompBoost, DenseCompBoost, CmplxCompBoost*) for 10 iterations and use validation dataset to decide the best early stopping rule. For End-2-End training, we tune two values of depth: $5, 10$. Tables 8, 9 presents the list of all the other hyper-parameters tuned.

Table 5: Details of simulated datasets used in our experiments. We use $20\%$ of the training data as validation set for picking the best hyper-parameter

| Dataset | Simulation 1 | Simulation 2 | Simulation 3 |
|---|---|---|---|
| # Train samples | 1000000 | 1000000 | 1000000 |
| # Test samples | 500000 | 500000 | 500000 |
| # Classes | 2 | 2 | 2 |

Table 6: Details of benchmark datasets used in our experiments. We use $20\%$ of the training data as validation set for picking the best hyper-parameter

| Details | Image Datasets | | | | |
|---|---|---|---|---|---|
| | SVHN | FashionMNIST | CIFAR10 | Convex | MNIST-rot-back-image |
| # Train samples | 73257 | 60000 | 50000 | 8000 | 12000 |
| # Test samples | 26032 | 10000 | 10000 | 50000 | 50000 |
| # Classes | 10 | 10 | 10 | 2 | 10 |

| Details | Tabular Datasets | | | |
|---|---|---|---|---|
| | MNIST | Letter | CovType | Connect4 |
| # Train samples | 60000 | 15000 | 464809 | 54045 |
| # Test samples | 10000 | 5000 | 116203 | 13512 |
| # Classes | 10 | 26 | 7 | 3 |

Table 7: List of hyper-parameters tuned for XGBoost, on all the datasets used in our experiments.

| Parameter | Values Tuned |
|---|---|
| Tree Depth | $\{10, 15, 20\}$ |
| Learning Rate | $\{0.1, 0.2\}$ |
| Number of Trees | $\{400, 800, 1600\}$ |

Table 8: List of hyper-parameters tuned for various compositional boosting techniques and end-2-end training.

| Dataset | Hyper-parameters tuned |
|---|---|
| Simulation-1 | width:$\{32, 64, 128\}$ |
| Simulation-2 | width:$\{64, 128, 256\}$ |
| Simulation-3 | width:$\{256, 512, 1024\}$ |
| SVHN | output channels:$\{32, 64, 128\}$ |
| FashionMNIST | output channels:$\{32, 64, 128\}$ |
| Convex | output channels:$\{32, 64, 128\}$ |
| MNIST-rot-back-image | output channels:$\{32, 64, 128\}$ |
| CIFAR10 | output channels:$\{32, 64, 128\}$ |
| MNIST | width:$\{256, 512, 1024\}$ |
| LETTER | width:$\{256, 512, 1024\}$ |
| Covtype | width:$\{1024, 2048, 4096\}$ |
| Connect4 | width:$\{256, 512, 1024\}$ |

Table 9: List of hyper-parameters tuned for AdaBoost and additive feature boosting. To be fair for additive boosting techniques, we considered wider weak learners than the ones used for compositional boosting and end-2-end training.

| Dataset | Hyper-parameters tuned |
|---|---|
| Simulation-1 | width:$\{256, 512, 1024\}$ |
| Simulation-2 | width:$\{256, 512, 1024\}$ |
| Simulation-3 | width:$\{4096, 8192, 16384\}$ |
| SVHN | output channels:$\{128, 256, 350, 512\}$ |
| FashionMNIST | output channels:$\{128, 256, 350, 512\}$ |
| Convex | output channels:$\{128, 256, 350, 512\}$ |
| MNIST-rot-back-image | output channels:$\{128, 256, 350, 512\}$ |
| CIFAR10 | output channels:$\{128, 256, 350, 512\}$ |
| MNIST | width:$\{256, 512, 1024\}$ |
| LETTER | width:$\{256, 512, 1024\}$ |
| Covtype | width:$\{4096, 8192, 16384\}$ |
| Connect4 | width:$\{256, 512, 1024\}$ |