[Reviews · NeurIPS 2020]

Review 1

Summary and Contributions: This paper proposes an extension of the popular boosting idea: building a strong ensemble learner in a greedy manner using a sequence of weak learners. Traditional boosting machines operate in the label space: each weak learner is trained to predict the cumulative residual error from the previous iterates. Furthermore, the resulting ensembles are always additive and prediction is performed as a simple linear combination of all component models. The paper argues that this approach works well in application domains with tabular datasets with well-engineered features, but it tends to perform badly in domains like image classification with complex decision boundaries and low-level features (e.g. pixels). This paper proposes to solve this problem by (a) boosting in the representation space (e.g. using feature transformers) rather than the label space and (b) allowing more complex aggregation of weak learners rather than simple addition (e.g. functional composition). The paper argues that, for a given choice of the transformer class \mathcal{G}, this idea is equivalent to existing approaches for layer-by-layer training of neural networks. The paper proposes two different ways to choose the transformer class at each iteration: DenseCompBoost and CmplxCompBoost. The former selects a function $g$ composed with a linear combination of all previous transformers (something akin to a DenseNet architecture) and the latter chooses a transformer class that explicitly increases in complexity at each iteration (e.g. wider and wider neural networks). The authors argue that these new greedy approaches overcome various problems which affect existing layer-by-layer methods (e.g. error propagation) and thus should allow one to better approach the accuracy of end-to-end training. The paper also presents excess risk bounds that depend on the Rademacher complexity of the transformer class at each boosting iteration. Experimental results are presented using synthetic, image and tabular data to argue that the two proposed schemes (DenseCompBoost and CmplxCompBoost) achieve better accuracy that (a) tree-based boosting, (b) additive boosting (whether in label or representation space) and (c) layer-by-layer training.

Strengths: This is a dense paper with a lot of ideas and content. The work is certainly relevant to the NeurIPS community since it goes some way to bridging the gap between two of the most popular ML subfields today boosting machines (widely used on tabular datasets) and neural networks (widely used on image/audio/text). The core ideas, as stated above, are elegant from a theoretical perspective, and the paper reveals insightful connections with existing approaches for layer-by-layer training of neural networks.

Weaknesses: When reading the main manuscript, some serious concerns arise regarding the experimental section. For instance, it is stated that XGBoost using decision stumps (e.g. depth 1) and it is not clear whether hyper-parameter tuning of the different schemes was performed. After consulting the supplemental material, these concerns are mostly addressed (e.g. hyper-parameters were tuned, XGBoost could use trees up to depth 20). However, these things need to be made clearer. In Section 3, two different algorithms are proposed to solve the optimization at each generalized boosting iteration. Algorithm 2 solves the problem exactly, and Algorithm 3 uses a first-order approximation akin to the gradient boosting of Friedman in the standard boosting setup. In Section 4, excess risk bounds are derived for the gradient-based method (Algorithm 3) and it is then stated as a corollary that the same bounds apply to Algorithm 2. However, there appears to be no experimental comparison between Algorithm 2 and Algorithm 3. The authors state that Algorithm 3 may be useful if the feature transformations are non-differentiable (e.g. if one uses trees) but that doesn't seem to be the case of interest. The paper is already quite dense and may benefit from being streamlined to focus solely on Algorithm 2.

Correctness: While I don't object to any of the claims in the main manuscript, the theoretical claims are presented without any sketch proof or argument. I have not checked the correctness of the proofs in the supplemental material. It would be easier to assess if a brief sketch of the steps of the proof were included in the main manuscript.

Clarity: The paper is written to a high standard but some sections feel a bit over-compressed. In particular, the text in the experimental section is hard to parse and the explanation of the different methods under study could be structured more clearly.

Relation to Prior Work: The paper clearly places the work in the context of the boosting literature, as well as the neural network literature.

Reproducibility: Yes

Additional Feedback: Minor comments: - Table 1: the "half width" row is described only in the caption but nowhere in the text. It is not clear what this adds to the message. === Update after author response === I thank the authors for their response. I believe this paper should be accepted and presented at NeurIPS.


Review 2

Summary and Contributions: The paper proposes a new framework for boosting models in the representation space, i.e. by combining weak feature extractors and learning a linear predictor on top of the combination rather than combining a set of weak predictors. In particular, this framework can be used to train neural networks in a greedy layer-by-layer manner. Excess risk bounds are derived for it and an empirical evaluation is performed on tabular and image datasets, comparing the framework instantiated with two choices of weak learner composition to state-of-the art greedy optimization techniques and end-to-end training of NNs. ---- UPDATE ----- After reading the other reviews and the rebuttal, I am still concerned by the limited novelty of the contributions, but mostly by the weakness of the empirical analysis. In the main text, empirical evidence should be reported to show in which cases and why the proposed incremental optimization should be preferred over end-to-end training of nns or boosting of weak learners. Overall, I still think that the contributions are significant (and because of this, I increased my score). However, the paper should be revised to become a clear accept.

Strengths: The proposed general framework for boosting weak feature extractors and its theoretical analysis is significant, as it defines a notion of weak learner in this context and derives generic excess risk bounds to study the expected performance of its different instantiations (choice of composition, set of weak learners). Moreover, the proposed methodology for building and training neural networks is shown to perform significantly better than existing incremental optimization techniques on the selected tabular and image datasets.

Weaknesses: The novelty of the contributions seem limited. Training neural networks with boosting has already been proposed in the past, as the paper highlights, but also the function set and proposed compositions are not significantly novel as they are very similar to ResNet's and DenseNet's skip connections. The empirical analysis misses some elements for assessing the significance of the method. For starting, execution times should be also reported as the principal reason for incrementally training NNs is to speed-up optimization. Also, the number of iterations of boosting (hence the size of the final networks) should be reported along with a study of the connections and type of layers that are used in practice by the network to make its decisions. This would allow the reader to understand which elements are actually key in improving the performance with respect to state-of-the-art boosting techniques. I also believe that more challenging datasets should be used in the experiments, especially those where standard NNs are known to be less performant, in order to understand the real impact of the methodology. For instance, one could consider the tabular datasets [1] and [2]. [1] https://archive.ics.uci.edu/ml/datasets/yearpredictionmsd [2] https://www.csie.ntu.edu.tw/ cjlin/libsvmtools/datasets/binary.html

Correctness: The theoretical analysis seems sound. The empirical evaluation is insufficient to understand why the proposed methodology performs better than state-of-the-art techniques. On a side note, decision trees can be differentiable (as opposed to what stated in page 4 of the paper). See e.g. [3]. [3] Popov, Sergei, Stanislav Morozov, and Artem Babenko. "Neural Oblivious Decision Ensembles for Deep Learning on Tabular Data." arXiv preprint arXiv:1909.06312 (2019)

Clarity: The paper is generally well written and easy to read. More details on the update step of Algorithms 3 could be reported, in particular to highlight its differences with a L1-constrained Frank Wolf optimization step.

Relation to Prior Work: To my knowledge, the related work is well reported and compared with.

Reproducibility: No

Additional Feedback:


Review 3

Summary and Contributions: The authors devise a generalized framework for boosting weak learners to a strong learner. On the one hand, the authors integrate function composition into the aggregation. The alternative choices of the update algorithms show superior performances compared with existing layer-by-layer training techniques. On the other hand, the generalized boosting framework could achieve lower error bound.

Strengths: The paper is well-written and easy to follow. Also the paper is well supported by theoretical analysis. The experiments in the paper and supplementary show some merit of the method.

Weaknesses: 1. The title of this paper is inappropriate. First, there is some papers also named in similar manner: [a] Julio Cesar Duarte, Ruy Luiz Milidiu. Generalized Boosting Learning. 2007. http://www.dbd.puc-rio.br/depto_informatica/07_15_duarte.pdf [b] Sonia Amodio. Generalized Boosted Additive Models. 2011. http://www.fedoa.unina.it/8696/1/amodio_sonia_23.pdf But unfortunately, authors did not cite these papers, and did not discuss the difference against them. Second, authors should highlight and clarify how the boosting is generalized or for what it has to be generalized in the title, e.g., generalizing weak learner or generalizing the additive combination mechanism? 2. I think it is better to integrate Algorithm 1,2 and 3 into a single algorithm. 3. I remain concerned about the meaning of the method, since the results are always lower than the end-to-end network. 4. Many popular greedy algorithms for learning deep neural networks (DNNs) can be derived from the framework. Could the method contribute to the optimization of DNNs? I think this part could be discussed in Broader Impact. 5. It would be better to visualize some of the learned weak learners for better understanding of the algorithm merit. 6. The experimental result is not supportive to the claim in this paper. Authors say that their method has similar performance to that of DNN, but why there is no DNN method (such as the refered ResNet) compared in the experiment? ------------ The reviewer have read the author response and comments from other reviewers. This paper has some merits by building the connection between deep learning and boosting, but there are also several major concerns, e.g., on the experimental details and connection to existing work, I stick to my original score.

Correctness: The methods and equations seem correct for me. In more detail, I wonder if some regularization on W (e.g., L1 or L2 -norm) should be needed to avoid over-fitting. Please also see weakness for more detailed comments.

Clarity: This paper is well organized.

Relation to Prior Work: Some discussion has been provided, but it seems that the difference is not well demonstrated. More discussion, better with some graphical illustration, is encouraged.

Reproducibility: Yes

Additional Feedback: I prefer to see more discussions on the effect of the method.


Review 4

Summary and Contributions: Instead of learning an end-to-end model, the paper proposes to learn new representations using deep learning, and put them into boosting. The learning algorithm has a performance close to the end-to-end deep learning networks, but also has theoretical guarantees. The paper focuses on function composition and on complex forms of aggregation of weak classifiers, namely, using deep learning models. UPDATE: the authors answered my questions (but not extremely precisely). I still think that the paper is very dense; is on the border between two highly popular topics (boosting and deep learning), however, the novelty of the paper is limited.

Strengths: The paper proposes a model that seems to function well in practice, and the theoretical guarantees for the algorithm are also provided.

Weaknesses: Although the results are interesting, the proposed model seems to be quite complex, since the weak classifiers are quite computationally expensive. I wonder what is computationally more effective: an end-to-end approach or the proposed one? Another obvious weakness of the proposed method is that its empirical performance is inferior to the end-to-end approaches. It would be interesting to focus on applications where the proposed method performs better, and to discuss why.

Correctness: The results seem to be sound. It can be that I missed it, however, to have the theoretical guarantees mentioned in the paper, it is necessary to be sure that the deep learning performs optimally?

Clarity: In general, the paper is well-written. However, there are some paragraphs that are not easy to read and, I would even say that the paper is not self-contained. So, to understand Proposition 3.1, it is expected that a reader knows well the work of Huang et al., ResNet, and the method of Bengio and al. Why a typical choice for \phi is a neural network (Line 107)? In the experimental section, line 308, it is mentioned that XGBoost uses decision stumps as weak classifiers. Why? The XGBoost is not limited to the decision stumps only.

Relation to Prior Work: The relation to the prior work is well explained.

Reproducibility: Yes

Additional Feedback:

[Author Response · NeurIPS 2020]

We thank the reviewers for their feedback. We would like to note that due to space constraints, we relegated some theoretical and experimental details to the Appendix. We tried to ensure that the main paper, together with appendix, is self-contained. In Appendix B, we describe the greedy algorithms of Huang et al and Bengio et al. Appendix D contains a discussion of the theoretical results. Appendix J contains the experimental setup, hyper-parameter selection. In the final version of the paper, we will move additional clarifying details from the appendix to the main paper, as detailed in our responses below. Regarding XGBoost experiments, line 308 has a typo. We actually used decision trees (not decision stumps) in our experiments. The list of hyper-parameters tuned for XGBoost can be found in Table 7 in the appendix. We used trees with depth up to 20 in our experiments.

**Reviewer 1.** *Algorithms 2, 3.* We believe having both the algorithms can provide a unified view of various boosting algorithms: many classical boosting algorithms can be derived using Algorithm 3; whereas, greedy algorithms for DNN training can be derived using Algorithm 2. Due to space constraints, we didn't add experimental results comparing both the algorithms. In our experiments, we noticed Algorithm 3 has marginally worse performance than Algorithm 2.

**Reviewer 2.** *Novelty.* As pointed out in Section 3.1 and Appendix J.1, existing greedy techniques have certain drawbacks which cause them to perform poorly. One of the contributions of our work is to fix these drawbacks using better greedy techniques. While the resulting architecture might resemble ResNets, DenseNets, they have certain crucial differences which cause them to perform better.

*Training NNs with boosting has already been proposed.* Proposition 3.1 proves the non-obvious result that all these techniques are actually equivalent to greedy layer-by-layer technique of Bengio et al and hence face the drawbacks stated in Section 3.1. Moreover, our proposed techniques strictly improve upon the existing techniques.

*Goal of our work.* We would like to emphasize that the reason behind studying greedy algorithms is not to speed-up optimization of deep networks. Rather, the main reason is to come up with algorithms which have strong theoretical guarantees and at the same time have good empirical performance. Our generalized framework satisfies both the criteria (also see our response to Reviewer 3). Moreover, greedy algorithms come with certain additional advantages such as low memory requirements. For details on computation time, see our response to Reviewer 4.

*Empirical Evaluation.* Details about the hyper-parameters such as number of boosting iterations can be found in Appendix J.2. We used a hold-out set to choose the number of boosting iterations, with an upper limit of 15 iterations. For all the datasets, the optimal iteration picked using this process is less than 15. Lines 315-317 specify the types of weak feature transformers used. In Appendix J.1, we provide more experimental results showing why the proposed techniques work better than StdCompBoost. We believe our results clearly show that the proposed techniques achieve the above stated goals. On almost all the datasets, our techniques are better than StdCompBoost and other additive boosting techniques. On some datasets, we even match the performance of end-to-end training. Finally, thanks for references [1,2]; we will add more experimental results using datasets from these repositories.

**Reviewer 3.** *Related Work.* Thanks for pointing out [a,b], which we were unaware of. In [a], the authors propose a slight variant of AdaBoost in which the initial distribution is allowed to be any probability distribution, and the classification errors are measured using a general cost function. In [b], the authors consider an additive boosting procedure where different kinds of weak hypothesis classes are used in each iteration. Our work differs from [a,b] in a number of aspects, the key aspect being the way we aggregate weak classifiers. [a,b] consider additive combinations of weak classifiers, whereas we consider more general combinations. Moreover, unlike [a,b], the complexity of weak hypothesis classes in our algorithms grows with iteration.

*Goal of our work.* Our framework is a first step towards bridging the gap between classical methods such as boosting and modern deep learning methods. Classical boosting comes with strong theoretical guarantees, but doesn't match the performance of DNNs. Whereas, DNNs have good empirical performance, but don't come up with strong theoretical guarantees. Our proposed framework allows to derive learning algorithms that come with strong theoretical guarantees and improve the performance of existing greedy algorithms (see conclusion).

*Comparison with end-to-end.* In our experiments we did compare with ResNets: end-to-end training corresponds to ResNets trained using SGD. We would like to emphasize that our goal is not to beat end-to-end trained deep networks (see our claims in abstract lines 16-20 and introduction lines 65-70). While matching the performance of end-to-end training is the ideal goal, in this work we set out with a different goal: we aim to improve the performance of existing greedy techniques and take a step towards bridging the performance between greedy algorithms and end-to-end training.

**Reviewer 4.** *Theoretical guarantees.* For the theoretical guarantees to hold, we only need the feature transformers that are fit at each iteration to satisfy certain weak learning condition (Definition 4.1). This condition doesn't require the shallow networks to be trained optimally. This is similar to the weak learning condition in traditional boosting, which requires the weak learners added at each iteration to have better than random performance.

*Computation.* In terms of training time, we didn't notice any significant advantage for greedy techniques over end-to-end training. However, greedy techniques consume very little memory compared to end-to-end training. So greedy techniques can be very useful for training huge models on GPUs with very little memory. That being said, greedy techniques has several other advantages: they are easy to optimize and are theoretically much easier to analyze, as one only needs to understand optimization of shallow networks.

[Meta-Review · NeurIPS 2020]

This paper presents a boosting variant that outperforms layer wise training of a neural network, while being quite similar conceptually. The paper is a good mix of theory and empirical evaluation, showing good results compared to both traditional boosting and layer wise training. It cannot beat end-to-end training, but needs a lot less memory, as no back propagation is needed. This makes it a valuable contribution in my opinion.